# ShapCCS: Shapley-Driven Client Coreset Selection in Federated Learning

**Shuo Ji** [1] **Jie Hu** [2] **Zhouqiao He** [2] **Zijie Zhao** [2] **Tianrui Li** [2] **Jie Xu** [3]

## Abstract

Computation overhead has emerged as a critical bottleneck in Federated Learning (FL). Coreset selection tackles this challenge by constructing an informative subset to represent the full dataset. However, existing approaches optimize coreset construction solely at the data level and enforce a uniform retention ratio across all clients, ignoring client heterogeneity and introducing detrimental fragmented clients. In this paper, we first introduce a *gradient projection Shapley value* (GPSV) to evaluate client contributions. GPSV captures both the directional and magnitude information of client updates and enables exact Shapley value calculation with $\mathcal{O}(1)$ per-coalition evaluation. Building on GPSV, we then propose ShapCCS, the first client-level coreset selection strategy for FL. ShapCCS prioritizes clients with high GPSV scores while excluding fragmented clients with negligible or even negative GPSV. As a client-level coreset selection strategy, ShapCCS can be integrated with a data-level selection approach, and additionally reduces communication costs, an advantage unattainable by data-level methods alone. Extensive experiments demonstrate the superiority of ShapCCS on model performance and robustness to noise. The code is available at https://github.com/iz70778/ShapCCS.

## 1. Introduction

Federated Learning (FL) (McMahan et al., 2023) is a distributed machine learning paradigm that enables multiple clients to collaboratively train a shared global model by synchronizing parameters with a central server while keeping raw data on local devices (Wen et al., 2023). FL has

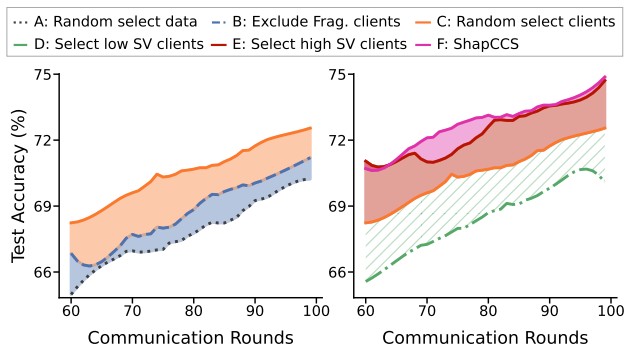

[1]SWJTU-Leeds Joint School, Southwest Jiaotong University, Chengdu, China [2]School of Computing and Artificial Intelligence, Southwest Jiaotong University, Chengdu, China [3]School of Computer Science, University of Leeds, Leeds, United Kingdom. Correspondence to: Tianrui Li <trli@swjtu.edu.cn>.

*Proceedings of the 43rd International Conference on Machine Learning*, Seoul, South Korea. PMLR 306, 2026. Copyright 2026 by the author(s).

*Figure 1.* Performance of different coreset selection strategies on CIFAR-10 with a 70% coreset budget. (a) Building upon A with random data selection, B further excludes 35% fragmented clients yet attains higher accuracy, indicating the adverse effect of such clients. C performs a random client-level coreset selection, completely eliminating fragmented clients and further improving performance. (b) With C as a reference, D retains low SV clients and suffers a notable performance drop, whereas E prioritizes high SV clients and yields a significant accuracy improvement. F (ShapCCS) augments E with a data-level coreset selection method, Moderate (Xia et al., 2023), resulting in additional gains.

extensive applications in privacy-sensitive fields, including medical care (Kaissis et al., 2020), finance (Chatterjee et al., 2024), and autonomous driving (Li et al., 2022). Despite privacy-preserving advantages, FL faces a computational bottleneck as model complexity, dataset volume, and the number of clients scale up (Pfeiffer et al., 2023).

Coreset selection, a classical technique in machine learning, reduces computational burden by identifying a representative subset of training data, such that a model trained on it achieves accuracy comparable to that on the full dataset (Toneva et al., 2019). However, existing coreset selection approaches are primarily designed for centralized settings and cannot be directly applied to FL (Hao et al., 2025), where data is distributed across clients and exhibits severe Non-IID characteristics (Liu et al., 2025). Although recent studies have explored coreset selection in FL (Hao et al., 2025; Sivasubramanian et al., 2024), they typically consider the coreset formation only at the data level and impose a uniform selection ratio among all clients, leading to two critical problems:

**(1) Introducing fragmented clients.** In Figure 1 (a), we first randomly construct a coreset containing 70% training data (curve A), and then further exclude 35% clients with the least data (curve B) while keeping the per-round client participation ratio unchanged. Notably, these removed clients account for only 13% of the whole data, and their exclusion results in an accuracy improvement rather than a degradation. We refer to such clients with minimal data as fragmented clients, by analogy to memory fragmentation in operating systems: both are sparse, marginal, and detrimental to the system. Beyond the accuracy gain and an additional 13% reduction in computation, the removal of fragmented clients also saves 35% communication cost since fewer participants per round entail less model parameter transmission. This observation motivates a paradigm shift from data-level coreset selection, which constructs a coreset by sampling individual data points, to client-level coreset selection, which builds the coreset by selecting clients rather than data. As a result, each selected client retains the entire dataset after selection, thereby eliminating fragmentation and remarkably improving performance, as shown by curve C. Client-level coreset selection permanently prunes clients to reduce per-round computation. It differs from client selection works (Tian et al., 2024) that accelerate FL through biasing client participation while keeping both clients and data unchanged, resulting in no reduction in per-round computation overheads. We clarify the differences between the two tasks in Appendix A.

**(2) Ignoring client heterogeneity.** Prior works (Wu et al., 2026; 2024; Sun et al., 2023) have established that clients contribute differentially to FL. Shapley value (SV) (Kuhn & Tucker, 1953), a principled metric from cooperative game theory, has been broadly applied in FL for contribution evaluation (Lei et al., 2025), client incentive (Tastan et al., 2024), and knowledge distillation (Shadin & Zhang, 2025). As shown in Figure 1 (b), selecting clients with high SV (curve E) yields consistent accuracy improvements compared to random client coreset selection, whereas retaining clients with low SV (curve D) leads to notable performance decline. This observation motivates a paradigm shift from uniform ratio coreset selection to contribution-aware coreset selection, in which the data retention ratio for each client is assigned based on its contribution.

Concerning the two shifts elaborated above, we propose the Shapley-Driven Client Coreset Selection (ShapCCS), a plug-and-play method that can integrate with a data-level coreset selection strategy $\mathcal{P}$. Specifically, we first design a *gradient projection Shapley value* (GPSV) that obviates the validation set required by accuracy-based SV (Liu et al., 2022), and incorporates magnitude information absent from gradient-based SV (Xu et al., 2021). By leveraging projection additivity and Gray Code enumeration, GPSV optimizes the per-coalition utility computation to $\mathcal{O}(1)$ without

compromising client interaction effects. Based on GPSV, ShapCCS allocates selection budgets proportionally to this score, excludes fragmented clients with budgets below the threshold $\tau$, and applies a data-level selection method $\mathcal{P}$ to further prune redundant data within the retained clients, yielding the accuracy gains shown by curve F in Figure 1 (b).

ShapCCS achieves a fragmentation-free and contribution-aware client-level coreset selection, simultaneously reducing the computational and communication costs. Theoretical analysis reveals that ShapCCS guarantees a bounded performance after selection. Experimental results demonstrate that ShapCCS consistently outperforms data-level coreset selection methods on performance and noise robustness. We summarise our main contributions as follows:

- We design a *gradient projection Shapley value* (GPSV) that captures both gradient alignment and magnitude. To the best of our knowledge, GPSV is the first SV algorithm to reduce per-coalition evaluation to $\mathcal{O}(1)$, enabling exact calculation in most scenarios.

- Based on GPSV, we propose ShapCCS, which is the first client-level coreset selection approach in FL and can be combined with a data-level coreset selection method. ShapCCS prioritizes high GPSV participants while excluding fragmented clients, effectively reducing both computation and communication overhead.

- Theoretical analysis and extensive experiments demonstrate that ShapCCS maintains model performance after selection and outperforms existing data-level coreset selection approaches.

## 2. Related work

### 2.1. Coreset Selection

Previous research on reducing the computational cost of FL has primarily focused on model pruning (Jiang et al., 2023; Qiu et al., 2022) and client selection (Yan et al., 2023; Tang et al., 2022). While coreset selection offers a promising alternative by pruning redundant samples to improve efficiency, its application in FL remains underexplored (Nagalapatti et al., 2022). This is mainly because current coreset selection methods are designed for centralized learning settings (Xia et al., 2023; Zheng et al., 2022) and typically prioritize hard samples (Paul et al., 2021; Pleiss et al., 2020). Directly applying these centralized approaches in FL can significantly decrease data coverage and exacerbate data heterogeneity, ultimately leading to severe client drift.

To alleviate the drift induced by pruning, GCFL (Sivasubramanian et al., 2024) constructs a coreset by matching the aggregated gradient with a reference gradient computed on a server-held validation set. However, this reliance on

a server-side validation dataset violates the privacy principles of FL. FedCS (Hao et al., 2025) applies a double pruning strategy that progressively removes samples around the decision boundary. Nevertheless, identifying boundary samples requires synchronization of class centers, which introduces additional communication costs. Furthermore, both approaches still fall into data-level coreset selection with a uniform retention ratio across all clients, ignoring client heterogeneity and introducing fragmented clients. To address these limitations, we propose ShapCCS, a client-level coreset selection framework that explicitly prioritizes high value participants and removes fragmented clients.

## 2.2. Shapley Value

Shapley value quantifies each player's contribution as the expected marginal gain from joining a coalition over all possible subsets of collaborators, which leads to an NP-hard computation (Wang et al., 2020). Existing studies generally approximate SV through Monte Carlo sampling (Song et al., 2019) and can be broadly categorized according to their utility definitions. One line of work defines utility as model accuracy and concentrates on enhancing sampling quality via subset truncation (Ghorbani & Zou, 2019; Wei et al., 2020), sample skipping (Zheng et al., 2023), and difference estimation (Sun et al., 2023). However, these methods inherently require an auxiliary validation dataset, which conflicts with the FL privacy constraints.

Another line of research (Xu et al., 2021; Tastan et al., 2024) defines utility as the cosine similarity between the coalition aggregated gradient and the global gradient, thereby eliminating the need for a validation set. Nevertheless, existing gradient-based approaches consider only directional alignment, whereas an effective update should progress not only in correct orientation but also with an appropriate step size (Ruder, 2016). An excessively large gradient update can induce oscillation and hinder convergence (Pan et al., 2025), while an overly small update may lead to stagnation in saddle points (Murata & Suzuki, 2022). To address this limitation, we propose GPSV, which adopts the gradient projection as utility to jointly encode both directional and magnitude information. It also optimizes the per-coalition evaluation complexity to $\mathcal{O}(1)$, thereby circumventing sampling approximation and enabling exact SV calculation.

## 3. Proposed Method

### 3.1. Federated Learning

We consider the standard FL setting with a client set $\mathcal{N}$, where each client $i \in \mathcal{N}$ possesses a local dataset $\mathcal{D}_i$ and a local loss function $F_i$. With $\mathcal{D} = \bigcup_{i \in \mathcal{N}} \mathcal{D}_i$, the global objective of FL is to find a model $\mathbf{w}^*$ that minimizes the

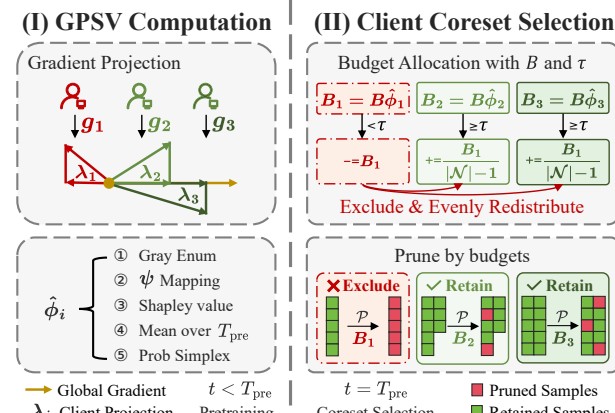

*Figure 2.* Overview of ShapCCS. (I) Pretraining stage ($t < T_{\text{pre}}$): each client is evaluated by its GPSV score $\hat{\phi}_i$. (II) Coreset selection stage ($t = T_{\text{pre}}$): the total selection budget $B$ is allocated proportionally to $\hat{\phi}_i$ to obtain client budgets $B_i$. Clients with $B_i < \tau$ are excluded, and their budgets are redistributed to remaining clients. Data within the retained clients are further pruned using a data-level coreset selection strategy $\mathcal{P}$ under budget $B_i$.

weighted average of local losses:

$$\mathbf{w}^* \in \arg\min_{\mathbf{w}} \sum_{i \in \mathcal{N}} \frac{|\mathcal{D}_i|}{|\mathcal{D}|} F_i(\mathbf{w}). \tag{1}$$

At each communication round $t = 0, \ldots, T - 1$, the server samples a subset $S_t \subseteq \mathcal{N}$ of size $\lceil \alpha |\mathcal{N}| \rceil$ with sampling ratio $\alpha \in (0, 1]$, and broadcasts the global model $\mathbf{w}^t$ to all clients in $S_t$. Each selected client $i \in S_t$ initializes its local model as $\mathbf{w}_i^{t,0} = \mathbf{w}^t$ and performs $K$ steps of local stochastic gradient descent to approach $\mathbf{w}_i^* \in \arg\min_{\mathbf{w}_i} F_i(\mathbf{w}_i)$:

$$\mathbf{w}_i^{t,k+1} = \mathbf{w}_i^{t,k} - \eta_t \nabla F_i(\mathbf{w}_i^{t,k}, \xi_i^{t,k}), \tag{2}$$

where $\eta_t$ is the learning rate and $\xi_i^{t,k} \sim \mathcal{D}_i$ is a stochastic mini-batch for $k = 0, \ldots, K - 1$. After local training, each client $i \in S_t$ uploads its local model $\mathbf{w}_i^{t,K}$ to the server, which computes the next round global model as:

$$\mathbf{w}^{t+1} = \sum_{i \in S_t} p_i^t \mathbf{w}_i^{t,K}, \tag{3}$$

where the aggregation weight $p_i^t = |\mathcal{D}_i| / \sum_{j \in S_t} |\mathcal{D}_j|$ is proportional to the data volume. For notational brevity, we denote the effective local gradient by $\mathbf{g}_i \triangleq \mathbf{w}^t - \mathbf{w}_i^{t,K}$ and the effective global gradient by $\mathbf{g} = \sum_{i \in S_t} p_i \mathbf{g}_i$, suppressing the round index $t$ throughout the paper.

Under the above FL setup, given a data retention ratio $r$ and a total selection budget $B = r|\mathcal{D}|$, ShapCCS aims to construct a client coreset $\mathcal{N}^+ \subseteq \mathcal{N}$ such that the aggregated

dataset $\mathcal{D}^+ = \bigcup_{i\in\mathcal{N}^+} \mathcal{D}_i$ satisfies $|\mathcal{D}^+| = B$. As illustrated in Figure 2, ShapCCS achieves this via a two-stage procedure. During the pretraining phase with $t < T_{\text{pre}}$, it utilizes GPSV to identify high-value participants and detect detrimental fragmented clients, as detailed in Section 3.2. At $t = T_{\text{pre}}$, it performs a client-level coreset selection by prioritizing clients with high GPSV scores, as described in Section 3.3. The pseudocode of ShapCCS is provided in Algorithm 1, where Lines 2–7 implement the standard FL process described in this section.

## 3.2. Gradient Projection Shapley Value

Given a set of players $S$ and a value function $\nu(\cdot)$, the Shapley value (SV) for player $i$ is defined as its average marginal contribution across all possible permutations of players. An equivalent expression is:

$$SV_i = \frac{1}{|S|} \sum_{C \subseteq S\setminus\{i\}} \frac{\nu(C \cup \{i\}) - \nu(C)}{\binom{|S|-1}{|C|}}. \qquad (4)$$

SV satisfies four desirable fairness properties: efficiency (total utility is fully distributed), symmetry (players with equal contributions receive equal SVs), linearity (SV is additive across games), and the null player property (players with no contribution receive zero SV).

Since gradient similarity based SV cannot capture the step size information, we define a novel value function $\nu$. It employs the projection length of a coalition's aggregated gradient onto the global update direction as utility, which jointly models both the alignment and magnitude of client updates.

**Definition 3.1** (Gradient Projection Value Function). Let $\lambda_i = \frac{\langle \mathbf{g}_i, \mathbf{g} \rangle}{\|\mathbf{g}\|^2}$ denote the projection of client $i$'s update and $\lambda_C = \frac{\langle \mathbf{g}_C, \mathbf{g} \rangle}{\|\mathbf{g}\|^2}$ be that of coalition $C$. By the additivity of projection, the utility of $C$ with the value function $\nu$ and dataset $\mathcal{D}_C = \bigcup_{i\in\mathcal{C}} \mathcal{D}_i$ can be defined as:

$$\nu(C) = \psi(\lambda_C) = \psi(\sum_{i\in C} \frac{|\mathcal{D}_i|}{|\mathcal{D}_C|}\lambda_i), \qquad (5)$$

where $\psi$ is a non-linear function designed to penalize excessively large gradients. Critically, the non-linearity of $\psi$ preserves the interaction effects of clients, thus preventing SV from degrading into individual marginal contributions, as shown in Theorem B.1. The specific form of $\psi$ is provided in Appendix B.2.

In practice, to mitigate the fluctuation of $\nu$ induced by gradient explosion or vanishing in deep neural networks, we compute $\lambda_i$ using the last layer gradients. This follows the empirical observation from Tastan et al. (2024) that the last layer signals are sufficiently informative, more robust,

and computationally efficient for contribution evaluation compared to the full model gradients (Xu et al., 2021).

Building on $\nu$, we propose the *gradient projection Shapley value* (GPSV) to quantify the partial contribution of each client to the global update.

**Definition 3.2** (Gradient projection Shapley value (GPSV)). Given the value function $\nu$, the GPSV of participant $i$ at round $t$ is defined as:

$$\phi_i^t = \frac{1}{|S_t|} \sum_{C \subseteq S_t\setminus\{i\}} \frac{\nu(C \cup \{i\}) - \nu(C)}{\binom{|S_t|-1}{|C|}}. \qquad (6)$$

To mitigate the evaluation bias arising from client selection, GPSV takes the mean value across $T_{\text{pre}}$ pretraining rounds:

$$\phi_i = \frac{1}{T_{\text{pre}}} \sum_{t=1}^{T_{\text{pre}}} \phi_i^t \qquad (7)$$

To facilitate budget allocation, we map GPSV to the probability simplex via non-negative shift and $\ell_1$-normalization.

$$\hat{\phi}_i = \frac{\phi_i - \min_{j\in\mathcal{N}} \phi_j}{\sum_{k\in\mathcal{N}} (\phi_k - \min_{j\in\mathcal{N}} \phi_j)}. \qquad (8)$$

Clients with large $\hat{\phi}_i$ provide updates closely aligned with the global gradient in both direction and magnitude, exhibiting the potential to recover the original update from these core clients. Conversely, small or even negative $\hat{\phi}_i$ suggest misaligned updates that may hinder convergence.

The pseudocode of GPSV is presented in Algorithm 1 (Lines 8–13). After computing the clients' gradient projection, it employs GrayEnum (Algorithm 2) to incrementally enumerate the projection of all coalitions. By the additivity of the projection utility (Def.3.1) and the structure of Gray Code, GPSV achieves $\mathcal{O}(1)$ per-coalition utility computation by only updating the single differing client between adjacent subsets. This stands in stark contrast to existing methods, which incur $\mathcal{O}(Vd)$ time for accuracy validation or $\mathcal{O}(d)$ for gradient similarity calculation, where $d$ is the model dimension and $V$ is the validation set size. Therefore, GPSV enables exact SV computation in cross-device settings with up to hundreds of participants. Even in massive scale scenarios, GPSV can also be combined with approximation strategies such as GTG truncation (Liu et al., 2022), yielding more accurate SV estimation by executing orders-of-magnitude more Monte Carlo samplings within the same time constraints.

## 3.3. Federated Learning with Client Coreset Selection

To eliminate fragmented clients while considering client heterogeneity, we propose the Shapley-Driven Client Coreset Selection (ShapCCS) with pseudocode in Algorithm 1

---

**Algorithm 1** Federated Learning with ShapCCS

---

1: **Input:** Client set $\mathcal{N}$, sampling ratio $\alpha$, total rounds $T$, pretraining rounds $T_{\text{pre}}$, total data retention ratio $r$, threshold $\tau$, data-level coreset selection method $\mathcal{P}$.

2: Initialize: $\mathcal{N}^+ \leftarrow \mathcal{N}$, total budget $B \leftarrow r|\mathcal{D}|$

3: **for** $t = 0$ **to** $T - 1$ **do**

4:      Sample $S_t \subseteq \mathcal{N}^+$ with $\lceil \alpha|\mathcal{N}^+| \rceil$

5:      Server broadcasts $\mathbf{w}^t$ to clients in $S_t$

6:      Local update: $\mathbf{g}_i \triangleq \mathbf{w}^t - \mathbf{w}_i^{t,K}, \ \forall i \in S_t$

7:      Aggregation: $\mathbf{w}^{t+1} \leftarrow \mathbf{w}^t - \sum_{i \in S_t} p_i \mathbf{g}_i$

8:      **if** $t < T_{\text{pre}}$ **then**

9:          $\mathbf{g}_i^\ell \leftarrow \text{LastLayer}(\mathbf{g}_i), \ \mathbf{g}^\ell \leftarrow \text{LastLayer}(\mathbf{g})$

10:          $\lambda_i \leftarrow \langle \mathbf{g}_i^\ell, \mathbf{g}^\ell \rangle / \|\mathbf{g}^\ell\|^2, \ \forall i \in S_t$

11:          $\{\lambda_C\}_{C \subseteq S_t} \leftarrow \text{GrayEnum}(\{\lambda_i\}_{i \in S_t}, \{p_i\}_{i \in S_t})$

12:          $\nu(C) \leftarrow \psi(\lambda_C), \ \forall C \subseteq S_t$

13:          $\phi_i \leftarrow \frac{1}{|S_t|} \sum_{C \subseteq S_t \setminus \{i\}} \frac{\nu(C \cup \{i\}) - \nu(C)}{\binom{|S_t| - 1}{|C|}}, \ \forall i \in S_t$

14:      **else if** $t = T_{\text{pre}}$ **then**

15:          Compute $\{\hat{\phi}_i\}_{i \in \mathcal{N}}$ via Eq 7 and Eq 8

16:          $\mathcal{N}^+ \leftarrow \emptyset$ and $B_i \leftarrow B\hat{\phi}_i, \ \forall i \in \mathcal{N}$

17:          **for** $i \in \mathcal{N}$ **in ascending order of** $\hat{\phi}_i$ **do**

18:              **if** $B_i < \tau$ and $\sum_{j>i} |D_j| \geq \sum_{j>i} B_j$ **then**

19:                  Redistribute $B_i$ evenly to $\{B_j \mid j > i\}$

20:              **else**

21:                  $\mathcal{N}^+ \leftarrow \mathcal{N}^+ \cup \{\mathcal{N}[i]\}$

22:              **end if**

23:          **end for**

24:          $D_i^+ \leftarrow \mathcal{P}(D_i, B_i), \ \forall i \in \mathcal{N}^+$

25:      **end if**

26: **end for**

---

(Lines 14–25). In Line 16, ShapCCS allocates total selection budget $B$ proportionally to GPSV calculated during pretraining phase. In Lines 17–23, it traverses clients in ascending order of GPSV, excludes fragmented clients with budgets below the threshold $\tau$, and redistributes their budgets to the subsequent clients. Eventually, in Line 24, ShapCCS applies a data-level coreset selection method $\mathcal{P}$ to further prune redundant samples for all retained clients.

Different from data-level coreset selection methods (Hao et al., 2025) that reduce only computation cost, ShapCCS simultaneously saves computational and communication overheads. After selection, the computation cost is reduced to $r = |\mathcal{D}^+|/|\mathcal{D}|$ of the original due to less data being processed. Meanwhile, the communication cost is reduced to $|\mathcal{N}^+|/|\mathcal{N}|$ of the original, since fewer clients participate

in training and transmit model parameters with the server.

The threshold $\tau$ controls the communication saving and client fragmentation by excluding fragmented clients and reallocating their budgets to high GPSV clients, thereby concentrating data on these informative participants. When $\tau = 0$, each client receives a budget exactly proportional to its GPSV without any redistribution from removed clients, leading to budget dispersion and introducing fragmented clients. In contrast, as $\tau \to \infty$, ShapCCS retains only the highest GPSV clients without applying $\mathcal{P}$, degenerating to curve E in Figure 1. We denote this special case as ShapCCS*, which maximizes communication savings by retaining only a few top GPSV clients and throughly eliminates fragmentation by keeping their entire dataset.

## 4. Convergence Analysis

Following Li et al. (2020), we make four standard assumptions, including $L$ smoothness, $\mu$ strong convexity, bounded gradient variance $\sigma^2$, and bounded gradient norm $G^2$. As in Cho et al. (2020), we quantify data heterogeneity using the Local-Global Objective Gap $\Gamma$ and Client Skew $\rho$. Formal statements of the above assumptions and definitions are deferred to Appendix C.1. Since ShapCCS evaluates clients via gradient projection, we define a gradient variant of $\rho$.

**Definition 4.1** (Gradient Client Skew). Let $\mathbf{w}'$ denote the model state at evaluation. The gradient client skew of $S$ is:

$$\rho_\nabla(S, \mathbf{w}') = \frac{\mathbb{E}_S[\sum_{i \in S} p_i \|\nabla F_i(\mathbf{w}')\|^2]}{\sum_{i \in \mathcal{N}} p_i(F_i(\mathbf{w}') - F_i(\mathbf{w}_i^*))}, \quad (9)$$

with $\bar{\rho}_\nabla \triangleq \min_{\mathbf{w}'} \rho_\nabla(S, \mathbf{w}')$ and $\widetilde{\rho}_\nabla \triangleq \max \rho_\nabla(S, \mathbf{w}^*)$.

We now present the convergence guarantee of ShapCCS as:

**Theorem 4.2.** *By setting* $\eta_t < \frac{1}{4L}$ *and* $\eta_t = \frac{\beta}{t+\gamma}$ *with* $\gamma > 0$ *and* $\beta > \frac{16}{\mu(8+3\bar{\rho}_\nabla)}$, *we have*

$$\mathbb{E}[F(\mathbf{w}^t)] - F^* \leq \underbrace{\frac{L(Q + 6\beta^2 L \Gamma \bar{\rho}_\nabla)}{2(t+\gamma)}}_{(I)} + \underbrace{\beta L \Gamma (\widetilde{\rho}_\nabla - \bar{\rho}_\nabla)}_{(II)},$$

$$(10)$$

*where* $Q = \gamma\|\mathbf{w}^0 - \mathbf{w}^*\|^2 + 32\beta^2 K^2 G^2 + \beta^2 \sigma^2$

Term (I) vanishes as $t \to \infty$ with decaying rate $\mathcal{O}(1/t)$. Term (II) is a non-vanishing bias induced by client-level coreset selection. In the case of full client participation and random selection, we have $\bar{\rho}_\nabla = \widetilde{\rho}_\nabla$ and term (II) vanishes, recovering the standard FedAvg. However, as the retention ratio $r$ diminishes, the aggregated stochastic gradients from selected clients suffer from severe variance inflation due to insufficient data convergence, which incurs the erratic fluctuations in $\rho_\nabla^+$, leading to the slow convergence when $\bar{\rho}_\nabla^+$ is excessively large, and the performance collapse when the bias $\widetilde{\rho}_\nabla^+ - \bar{\rho}_\nabla^+$ grows dominant.

*Table 1.* Average accuracies (%) on Fashion-MNIST, CINIC-10 and CIFAR-100 with Dir(0.1) and $r \in \{70\%, 50\%, 30\%\}$ over five trials. Full-dataset accuracies are 84.99±0.47, 52.91±0.37, and 42.44±0.85 respectively. Results exceeding the full dataset are marked with †.

| METHODS | FASHION-MNIST | | | CINIC-10 | | | CIFAR-100 | | |
|---|---|---|---|---|---|---|---|---|---|
| | 70% | 50% | 30% | 70% | 50% | 30% | 70% | 50% | 30% |
| RANDOM | 82.33±0.41 | 80.02±0.14 | 75.65±1.44 | 45.30±1.52 | 40.41±2.06 | 34.66±0.98 | 31.65±0.13 | 24.63±0.23 | 17.67±0.19 |
| CRAIG | 83.29±0.36 | 81.64±0.29 | 79.17±0.69 | 48.34±0.39 | 44.48±0.83 | 39.49±0.53 | 37.40±0.40 | 32.28±0.34 | 26.14±0.30 |
| EL2N | 84.03±0.58 | 82.90±0.35 | 81.32±0.37 | 49.74±0.51 | 47.73±0.82 | 43.71±0.71 | 37.12±0.46 | 33.08±0.37 | 28.57±0.34 |
| CCS | 84.04±0.37 | 82.88±0.51 | 81.25±0.16 | 49.79±0.53 | 47.17±0.57 | 44.01±0.86 | 37.64±0.63 | 33.12±0.48 | 28.36±0.22 |
| MODERATE | 83.74±0.56 | 82.75±0.55 | 81.04±0.48 | 50.02±0.18 | 46.76±0.40 | 43.80±0.37 | 37.22±0.47 | 33.05±0.34 | 28.15±0.36 |
| GCFL | 83.02±0.23 | 80.87±0.49 | 78.96±0.38 | 49.62±0.39 | 47.15±0.75 | 43.39±0.61 | 37.07±0.69 | 33.02±0.51 | 28.95±0.30 |
| FEDCS | 80.96±0.72 | 79.05±0.61 | 77.60±0.56 | 43.56±0.78 | 39.63±0.42 | 37.15±0.88 | 32.96±0.50 | 27.74±0.44 | 23.54±0.49 |
| S-FEDAVG | 83.46±0.50 | 81.54±0.54 | 77.71±1.33 | 48.51±0.83 | 46.36±0.84 | 39.41±2.02 | 38.72±0.69 | 33.92±0.75 | 26.85±0.83 |
| SHAPCSS* | 85.15±0.42† | **84.77±0.47** | 82.77±1.20 | 51.91±1.21 | 49.58±1.14 | 45.36±2.17 | **42.26±0.87** | 40.19±1.04 | **35.88±0.38** |
| SHAPCCS | **85.41±0.52†** | 84.66±0.64 | **83.25±1.63** | **52.60±1.34** | **49.65±1.62** | **46.97±3.24** | 41.69±0.82 | **40.19±0.81** | 35.79±0.75 |

ShapCCS alleviates this instability by selecting high GPSV clients to construct a client-level coreset whose $\rho_\nabla^+$ closely matches the original $\rho_\nabla$, hence achieving performance comparable to training on the complete dataset. To quantify the gap between $\rho_\nabla^+$ and $\rho_\nabla$, we further introduce two assumptions to model the subset utility approximation error and gradient misalignment.

**Assumption 4.3** (Subset Utility Approximation). $\nu(C)$, defined as the gradient projection followed by a nonlinear transformation $\psi$ (Def. 3.1), is approximately proportional to the fraction of the total Shapley value contributed by $C$:

$$\nu(C) = \left( \frac{\sum_{i \in C} \phi_i}{\sum_{i \in S} \phi_i} + \varepsilon \right) \|\mathbf{g}\|, \qquad (11)$$

where $\varepsilon$ denotes a bounded approximation error induced by the lack of utility additivity. When $\psi$ is the identity function, $\varepsilon$ becomes zero with GPSV degrading to individual marginal contribution.

**Assumption 4.4** (Gradient Misalignment). For each retained client $i$, the angle $\theta_i$ between its local gradient $\mathbf{g}_i$ and the global gradient $\mathbf{g}$ satisfies:

$$\cos \theta_i \leq \kappa, \qquad (12)$$

where $\kappa$ bounds the gradient discrepancy of selected clients.

We now bound the gap of $\rho_\nabla^+$ and $\rho_\nabla$ in Theorem 4.5.

**Theorem 4.5.** *Under Assumptions 4.3 and 4.4, $\rho_\nabla^+$ achieves the following approximation bounds:*

$$\left( \frac{\sum_{i \in S^+} \phi_i}{\sum_{i \in S} \phi_i} + \varepsilon \right)^2 \leq \frac{\rho_\nabla^+}{\rho_\nabla} \leq \frac{\kappa^2}{r}, \qquad (13)$$

The lower bound equals one when $|S^+| = |S|$, implying no coreset selection. As $|S^+|$ shrinks with selection, ShapCCS prevents the lower bound from degrading by preferentially retaining clients with high GPSV scores and accordingly maintains the lower bound close to one. On the other hand,

as $r$ decreases, ShapCCS tightens the upper bound by selecting clients with well aligned gradients, typically high GPSV clients with small $\theta$, which yields a smaller $\kappa$ that counteracts the expansion caused the by decline of $r$. Together, Theorems 4.2 and 4.5 establish that ShapCCS can construct a coreset with $\rho_\nabla^+$ close to the original $\rho_\nabla$, thus guaranteeing performance comparable to that of the full dataset.

## 5. Experiments

### 5.1. Experimental Setup

**Datasets.** We conduct comprehensive experiments on four commonly used datasets: Fashion-MNIST (Xiao et al., 2017), CINIC-10 (Darlow et al., 2018) CIFAR-10, and CIFAR-100 (Krizhevsky et al., 2009). In the IID setting, data are shuffled and assigned to clients such that the local dataset size follows a log-normal distribution with standard deviation 0.75. In the non-IID setting, data are partitioned accross clients using Dirichlet sampling with Dir(0.1) and Dir(0.5) concentration parameters.

**Baselines.** We compare ShapCCS and ShapCCS* with five coreset selection methods designed for centralized settings, including CRAIG (Mirzasoleiman et al., 2020), EL2N (Paul et al., 2021), CCS (Zheng et al., 2022), Moderate (Xia et al., 2023), and Random, and two approaches tailored for FL scenarios, including GCFL (Sivasubramanian et al., 2024) and FedCS (Hao et al., 2025). We also include a SV based client selection method, S-FedAvg (Nagalapatti & Narayanam, 2021), as a reference baseline, which is early stopped at $rT$ to ensure a same computational cost. We set the pretraining rounds to $T_{\text{pre}} = 0.1T$ and the number of clients to $|\mathcal{N}| = 100$ with a sampling ratio $\alpha = 15\%$. For all experiments, we report the mean accuracy over five random seeds.

**Implementation Details.** The data-level coreset selection strategy $\mathcal{P}$ of ShapCCS is set to Moderate (Xia et al., 2023), and the performance when combined with other methods is

*Table 2.* Average accuracies (%) on CIFAR-10 with $r \in \{70\%, 50\%, 30\%\}$ over five trials. Full-dataset accuracies under Dir(0.1), Dir(0.5), and IID are 61.51±0.80, 68.14±0.63, and 74.06±0.70, respectively. Results exceeding the full dataset are marked with †.

| METHODS | DIR(0.1) | | | DIR(0.5) | | | IID | | |
|---|---|---|---|---|---|---|---|---|---|
| | 70% | 50% | 30% | 70% | 50% | 30% | 70% | 50% | 30% |
| RANDOM | 52.98±1.12 | 46.43±3.38 | 40.43±2.18 | 55.96±1.44 | 44.79±0.84 | 39.67±1.30 | 70.83±0.65 | 67.60±1.02 | 62.85±1.34 |
| CRAIG | 56.17±0.83 | 51.49±1.25 | 45.88±1.42 | 63.39±0.53 | 58.12±0.87 | 51.27±0.82 | 69.50±1.01 | 65.08±1.43 | 58.75±1.54 |
| EL2N | 58.42±1.13 | 54.45±1.18 | 50.63±1.36 | 64.01±0.84 | 60.96±0.77 | 56.12±0.94 | 70.61±0.59 | 67.48±0.97 | 62.83±1.13 |
| CCS | 57.87±1.41 | 54.87±0.54 | 50.47±1.17 | 64.22±0.49 | 61.15±0.71 | 56.09±0.74 | 70.72±0.80 | 67.46±0.95 | 62.80±1.64 |
| MODERATE | 58.38±1.05 | 55.56±0.79 | 50.47±1.23 | 64.21±0.57 | 61.17±1.04 | 56.24±1.20 | 70.68±0.64 | 67.31±1.00 | 62.75±1.05 |
| GCFL | 58.71±1.28 | 54.90±0.61 | 50.37±1.30 | 63.93±0.59 | 59.12±0.45 | 53.90±0.82 | 69.70±0.87 | 65.53±0.96 | 60.31±1.61 |
| FEDCS | 50.63±1.10 | 45.61±1.22 | 42.90±1.40 | 58.33±1.17 | 53.31±0.94 | 48.40±0.89 | 67.16±1.06 | 63.56±1.10 | 58.88±1.25 |
| S-FEDAVG | 58.59±1.37 | 52.43±1.44 | 47.35±1.55 | 64.27±0.44 | 59.53±1.21 | 52.86±0.62 | 71.50±0.91 | 67.58±1.44 | 61.61±2.02 |
| SHAPCSS* | 61.91±1.14† | 58.34±2.69 | **55.37**±3.68 | 67.98±1.18 | 67.13±0.98 | **65.61**±1.12 | 75.11±0.44† | **74.83**±0.85† | **73.08**±0.51 |
| SHAPCCS | **62.72**±1.22† | **60.25**±1.46 | 55.26±2.94 | **68.16**±0.31† | **67.82**±0.87 | 65.16±0.87 | 75.10±0.56† | 74.02±0.45 | 72.59±0.45 |

*Table 3.* Average communication savings (%) on four datasets with Dir(0.1) and $r \in \{70\%, 50\%, 30\%\}$ over five trials.

| $r$ | FASHION-MNIST | CINIC-10 | CIFAR-10 | CIFAR-100 |
|---|---|---|---|---|
| 70% | 53.4 | 56.2 | 54.4 | 41.4 |
| 50% | 70.2 | 70.2 | 71.6 | 63.2 |
| 30% | 86.2 | 85.6 | 85.4 | 80.2 |

*Table 4.* Average accuracy gains (%) when enhancing baseline methods by ShapCCS on CIFAR-10 with Dir(0.1) over five trials.

| + SHAPCCS | 70% | 50% | 30% |
|---|---|---|---|
| RANDOM | 62.22 (↑**9.24**) | 60.04 (↑**13.62**) | 55.66 (↑**15.22**) |
| CRAIG | 62.00 (↑**5.83**) | 59.67 (↑**8.18**) | 54.49 (↑**8.61**) |
| EL2N | 62.56 (↑**4.14**) | 60.36 (↑**5.91**) | 55.39 (↑**4.76**) |
| CCS | 62.66 (↑**4.79**) | 60.15 (↑**5.28**) | 54.43 (↑**3.96**) |
| MODERATE | 62.72 (↑**4.34**) | 60.25 (↑**4.69**) | 55.26 (↑**4.79**) |
| GCFL | 61.38 (↑**2.67**) | 60.64 (↑**5.74**) | 53.54 (↑**3.17**) |
| FEDCS | 60.77 (↑**10.14**) | 55.77 (↑**10.16**) | 51.09 (↑**8.19**) |

reported in Table 4. Given $|\mathcal{N}| = 100$ clients, the threshold $\tau$ is set to $B/70$ for $r = 70\%$ and $B/50$ for $r = 50\%$ respectively, corresponding to evenly distributing $B$ across the expected $r|\mathcal{N}|$ retained clients. When $r$ is further reduced to 30%, the budget available to each client becomes significantly limited, and we therefore increase $\tau$ to $B/20$ to filter out fragmented clients.

### 5.2. Main Results

**Performance Comparison.** The performance of all methods on four datasets is reported in Table 1 and Table 2. Overall, ShapCCS consistently outperforms all competing approaches across various settings and even surpasses the full-dataset accuracy in several cases. Moreover, the comparable performance of ShapCCS and ShapCCS* indicates that pure client-level coreset selection alone already suffices to outperform the state-of-the-art data-level methods. When compared with the coreset selection paradigm under same computational constraints, the inferiority of S-FedAvg suggests a limitation of client selection approaches: they reduce computation primarily through accelerating convergence. However, under limited resources, this often results in insufficient training rounds that induce poor convergence.

**Disscussion and analysis.** As shown in Table 2, on CIFAR-10 with Dir(0.1) and $r = 70\%$, ShapCCS achieves a 4.01% accuracy gain over the second best approach. This advantage expands as the retention ratio $r$ and data heterogeneity Dir(·) decrease, peaking at 9.74% in the IID case with $r = 30\%$. We attribute this improvement to two reasons. First, current approaches overlook client heterogeneity and enforce a uniform $r$ across clients, leading to budget dis-

crepancies and fragmented clients as $r$ decreases. ShapCCS addresses this problem by excluding fragmented clients and assigning high GPSV clients with larger budgets. Second, data-level coreset selection methods relies on representative samples to compress redundant data. This is effective in highly non-IID scenarios where the client data is skewed towards a few classes and exhibits substantial intra-class redundancy. However, in IID case, each client holds all classes of data, resulting in only a few samples per class and accordingly limited redundancy. Therefore, directly applying data-level coreset selection methods will reduce local data diversity and decrease model performance. ShapCCS solves this by employing the client-level coreset selection to ensure each selected client retains most of its data.

**Computation and communication savings.** Neglecting the computation overhead of coreset selection, all methods achieve a computational reduction of $1 - r$. However, as for communication, existing data-level coreset selection yields no savings. In contrast, ShapCCS achieves a communication reduction of $1 - |\mathcal{N}^+|/|\mathcal{N}|$ by removing fragmented clients and constructing a client-level coreset. As shown in Table 3, on CIFAR-10 with Dir(0.1) and $r = 70\%$, ShapCCS reduces computation by 30% and communication by 54.4%, while attaining 62.72% accuracy that surpasses 61.51% obtained using the full dataset.

**Modular Composability.** Table 4 shows that ShapCCS enhances baseline performance by 2.67% to 15.22%. When integrated with Random data selection, ShapCCS maintains

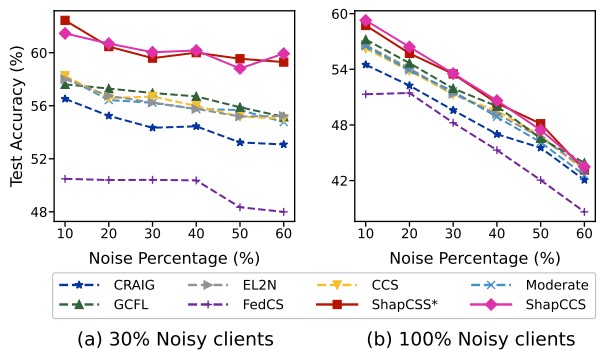

(a) 30% Noisy clients          (b) 100% Noisy clients

*Figure 3.* Robustness of ShapCCS under randomly flipped labels for 30% of clients in (a) and 100% of clients in (b) on CIFAR-10 with Dir(0.1) and $r = 70\%$ over five trials.

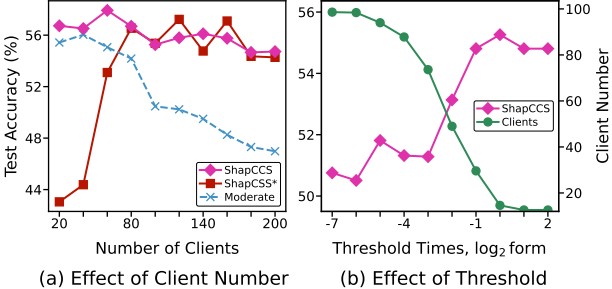

(a) Effect of Client Number          (b) Effect of Threshold

*Figure 4.* Hyperparameter effects. (a) Performance of ShapCCS under varying client scales. (b) Test accuracy and the number of retained clients under different multipliers of threshold where the x-axis is $\log_2$(multiplier). All experiments are conducted on CIFAR-10 with Dir(0.1) and $r = 30\%$ over five trials.

*Table 5.* Comparison of different Shapley value on CIFAR-10 with Dir(0.1) and $r \in \{70\%, 50\%, 30\%\}$ over five trials.

| $r$ | GTG-SV | NSV | CGSV | CSSV | GPSV |
|---|---|---|---|---|---|
| 30% | 60.34 | 61.11 | 60.54 | 59.73 | **61.91** |
| 70% | 57.21 | **59.48** | 58.66 | 56.33 | 58.34 |
| 50% | 54.49 | **55.38** | 54.37 | 49.76 | 55.37 |
| $\log_{10}(\text{Time (s)})$ | 2 | 3 | 1 | $-2$ | $-2$ |
| $\mathcal{O}(\cdot)$ | $\mathcal{O}(Vd)$ | $\mathcal{O}(Vd)$ | $\mathcal{O}(d)$ | $\mathcal{O}(1)$ | $\mathcal{O}(1)$ |

## 5.3. Noise Robustness

We evaluate the robustness of ShapCCS by randomly flipping $10\% - 60\%$ of labels for 30% and 100% of clients. As shown in Figure 3 (a), under 30% noisy clients, ShapCCS consistently outperforms all baselines and maintains high accuracy by filtering noisy clients with negative GPSV. In a more challenging setting with 100% noisy clients, ShapCCS undergoes a notable accuracy decline. This can be ascribed to its reliance on the global gradient as the evaluation reference. Under extreme noise conditions, the global gradient is thoroughly dominated by noisy updates, thereby inflating GPSV for malicious clients while underestimating scores for benign ones. Nevertheless, ShapCCS still slightly outperforms other methods, benefiting from the exclusion of fragmented clients. These results demonstrate the robustness of ShapCCS under different noisy scenarios.

## 5.4. Effect of hyperparameter

**Effect of client scales.** Figure 4 (a) depicts the accuracy of ShapCCS and ShapCCS* under varying client numbers $|\mathcal{N}|$, using Moderate(Xia et al., 2023) as a reference. When $|\mathcal{N}|$ is small, each client holds substantial data and is less likely to become fragmented, enabling Moderate to achieve accuracy competitive with ShapCCS. In contrast, ShapCCS* suffers a severe performance drop because the constrained total budget under $r = 30\%$, together with the large amount of data per client under small $|\mathcal{N}|$, restricts it to selecting only a few clients, which fails to adequately cover the data distribution under Dir(0.1), where each client contains data roughly from a single class. As $|\mathcal{N}|$ increases, the fragmented clients emerge due to the reduced amount of data per

client, leading to performance gains for ShapCCS* and the decline for Moderate. By comparison, ShapCCS maintains high accuracy across all $|\mathcal{N}|$, manifesting its effectiveness in both cross-silo and cross-device scenarios.

**Effect of threshold.** Figure 4 (b) shows the accuracy and number of retained clients of ShapCCS under different scales of threshold $\tau$. When $\tau$ is small, all clients are retained, leading to a significant performance drop due to the inclusion of fragmented clients. As $\tau$ increases, the performance improves and reaches its peak at $\tau = B/20$, suggesting the accuracy gain from threshold tuning. When $\tau$ becomes sufficiently large, ShapCCS degenerates into ShapCCS*, retaining only the top GPSV clients. In practice, ShapCCS* serves a simple yet effective default, while a properly tuned $\tau$ is preferred when fine-grained selection is required to maximize performance.

## 5.5. Comparison with other Shapley Value

We compare GPSV with four SV variants in the ShapCCS* framework, including two accuracy based methods, GTG-SV (Liu et al., 2022) and NSV (Sun et al., 2023), and two gradient based methods, CGSV (Xu et al., 2021) and CSSV (Tastan et al., 2024). Table 5 presents the accuracy along with the time scale and complexity of per-coalition evaluation. Overall, all variants achieve close accuracy. However, GTG-SV, NSV, and CGSV employ sampling approximation and incur substantial evaluation time, which even exceeds the model training time. Although CSSV

competitive accuracy, which indicates that ShapCCS is not restricted to the specific data selection methods. This in turn reveals a promising direction for designing a data-level selection strategy that is suitable for client-level coreset selection.

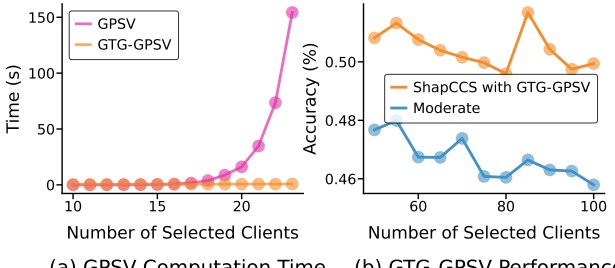

(a) GPSV Computation Time  (b) GTG-GPSV Performance

*Figure 5.* GPSV Estimation. (a) Computation time of exact GPSV and GTG-GPSV. (b) Test accuracy of ShapCCS when using GTG-GPSV. All experiments are conducted on CIFAR-10 with Dir(0.1) and $r = 30\%$ over five trials.

achieves a running time scale of $10^{-2}$, it directly adopts individual marginal contributions as SV and ignores client interaction effects, leading to the worst performance. In contrast, GPSV leverages gradient projection utility to incorporate the alignment and magnitude information of client gradients, attaining competitive accuracy while achieving a reduced computation time and $\mathcal{O}(1)$ complexity. The results show that GPSV provides a trade-off between evaluation quality and efficiency, offering a practical and effective contribution metric for client-level coreset selection.

### 5.6. Scalable GPSV Estimation

Although GPSV enables an $\mathcal{O}(1)$ per-coalition evaluation, exact GPSV computation still requires enumerating all $2^{|S_t|}$ subsets. This becomes a computational bottleneck as $|S_t|$ increases. However, GPSV is compatible with truncated estimation methods such as GTG-SV (Liu et al., 2022), which provide efficient SV approximation with limited accuracy loss. Specifically, we compute GPSV exactly up to $|S_t| = 20$. Beyond this threshold, we switch to GTG-based Monte Carlo estimation, denoted GTG-GPSV, which samples 1,000 permutations, resulting in at most $1,000 \times |S_t|$ subset evaluations per round. As shown in Figure 5, exact computation grows exponentially with $|S_t|$, whereas GTG-GPSV scales nearly linearly with a modest slope. Moreover, ShapCCS with GTG-GPSV still consistently outperforms Moderate (Xia et al., 2023), demonstrating its scalability to large-scale cross-device scenarios.

## 6. Conclusion

In this paper, we first introduced a *gradient projection Shapley value* with $\mathcal{O}(1)$ per-coalition evaluation to quantify client contributions. We then proposed ShapCCS, the first client-level coreset selection approach for FL. ShapCCS allocates selection budgets proportionally to the Shapley value and excludes fragmented clients, achieving a reduction in both computation and communication. Theoretical analysis guarantees the performance of ShapCCS. Experiments on

four datasets under IID and non-IID scenarios demonstrated its superior performance, modular composability, robustness to noise, and insensitivity to hyperparameters. Future works can explore data-level selection strategies that are suitable for client-level coreset selection.

## Acknowledgements

This research was supported by the National Natural Science Foundation of China (No. U2468207, 62576295), Sichuan Science and Technology Program (No. 2024NSFTD0036), Chengdu Science and Technology Program (No. 2024-YF05-00340-SN), and the Fundamental Research Funds for the Central Universities under Grant 2682026ZT007.

## Impact Statement

This paper proposes a Shapley-Driven Client Coreset Selection method for federated learning. It reduces both computational and communication overheads by prioritizing informative clients and excluding fragmented or detrimental ones. Our work aims to advance efficient and scalable federated learning. There are many potential societal consequences of our work, none which we feel must be specifically highlighted here.

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

## A. Difference Between Client Coreset Selection and Client Selection

Client selection and client coreset selection reduce computational cost through fundamentally different mechanisms, and should therefore be viewed as two distinct tasks. Given $S_t$ as the set of participating clients in round $t$, client selection ranks all available clients in every round and selects the top $|S_t|$ participants. As a result, it achieves no reduction in per-round computation and communication. The advantage of client selection lies in prioritizing high-value clients, thereby accelerating convergence and reducing the total number of communication rounds from $T$ to $T^+$. This indirectly results in a computation and communication reduction of $1 - T^+/T$.

In contrast, client coreset selection directly reduces the per-round computational cost itself. Given the full client set $\mathcal{N}$ and the entire dataset $\mathcal{D}$, client coreset selection constructs a subset of clients $\mathcal{N}^+ \subseteq \mathcal{N}$ with the retained dataset as $\mathcal{D}^+ = \bigcup_{i \in \mathcal{N}^+} \mathcal{D}_i$. This directly reduces per-round computation by $1 - |\mathcal{D}^+|/|\mathcal{D}|$ due to less data being processed within each round. Meanwhile, the per-round communication is reduced by $1 - |\mathcal{N}^+|/|\mathcal{N}|$ as fewer clients participate in training and exchange model updates with the server after selection.

Given a computation saving target $r$, coreset selection achieves this by building an informative subset $\mathcal{N}^+$ satisfying $|\mathcal{D}^+| = r|\mathcal{D}|$. In comparison, client selection typically meets the same computation constraint by reducing the number of training rounds to $T^+ = rT$, which may adversely affect convergence when $r$ is small. Therefore, in our experiments, we treat client selection as a reference method rather than a direct competitor, as the two approaches address different types of problems and are not directly comparable under the same computation constraint.

## B. Details of Gradient Projection Shapley Value

### B.1. Degradation of Shapley Value

We show that, under additive coalition utilities, Shapley value collapses to individual marginal contributions.

**Theorem B.1** (Degeneration of Shapley Value under Additive Utilities). *Let $S$ be a set of clients, and let the value function $v$ be additive, i.e. $v(C) = \sum_{i \in C} v(i), \forall C \subseteq S$. Then the Shapley value $\phi_i$ of each client $i \in S$ degrades to its marginal contribution:*

$$\phi_i(v) = v(i). \tag{14}$$

**Proof of Theorem B.1.** By definition, the SV of client $i$ is

$$\phi_i = \frac{1}{|S|} \sum_{C \subseteq S \setminus \{i\}} \frac{\nu(C \cup \{i\}) - \nu(C)}{\binom{|S|-1}{|C|}}. \tag{15}$$

Under the additivity assumption,

$$v(C \cup i) - v(C) = v(i), \quad \forall C \subseteq N \setminus \{i\}. \tag{16}$$

Hence the marginal contribution of client $i$ is independent of the coalition $C$. Factoring out $v(\{i\})$, we obtain

$$\phi_i(v) = v(i) \sum_{C \subseteq S \setminus \{i\}} \frac{1}{|S|\binom{|S|-1}{|C|}}. \tag{17}$$

The remaining summation equals 1, as can be verified by grouping subsets by cardinality:

$$\sum_{C \subseteq S \setminus \{i\}} \frac{1}{|S|\binom{|S|-1}{|C|}} = \sum_{k=0}^{|S|-1} \sum_{C \subseteq S \setminus \{i\}, |C|=k} \frac{1}{|S|\binom{|S|-1}{k}}$$

$$= \sum_{k=0}^{|S|-1} \binom{|S|-1}{k} \frac{1}{|S|\binom{|S|-1}{k}}$$

$$= 1. \tag{18}$$

Therefore,

$$\phi_i(v) = v(i), \tag{19}$$

Consequently, under additive utilities, SV assigns no credit to interactions among clients and reduces to individual marginal contributions. In such cases, SV offers no advantage over naive individual scoring and fails to reflect synergistic or antagonistic interactions among clients. To avoid this degeneration, we introduce a non-linear utility mapping $\psi$ that breaks additivity and preserves interaction effects.

### B.2. Design of $\psi$

We now provide the explicit form of function $\psi$, which is used in Def. 3.1 to convert the gradient projection length of a coalition into its corresponding utility. The design of $\psi$ is guided by three desiderata: (i) monotonic reward for informative and reasonably sized gradient contributions, (ii) tolerance plateau that preserves the utility of highly aligned but moderately over-scaled gradients, and (iii) penalization for excessively large gradient projections that may induce oscillation and impede convergence (Pan et al., 2025). For a coalition $C$, let $\lambda_C = \sum_{i \in C} \lambda_i$ denote the projection length of the coalition gradient onto the direction of global gradient $\mathbf{g}$. The utility mapping $\psi$ is:

$$\psi(\lambda_C) = \begin{cases} \lambda_C, & \text{if } \lambda_C \leq 1.25 \, |\mathbf{g}|, \\ 1.25 \, |\mathbf{g}|, & \text{if } 1.25 \, |\mathbf{g}| < \lambda_C \leq 1.75 \, |\mathbf{g}|, \\ 3 \, |\mathbf{g}| - \lambda_C, & \text{if } \lambda_C > 1.75 \, |\mathbf{g}|. \end{cases} \tag{20}$$

This design ensures that $\psi$ is computationally efficient, requiring only an arithmetic operation once the projection length $\lambda_C$ is computed. The specific numerical values (1.25, 1.75, and 3) define a moderately wide transition band between reward and penalty. Empirically, we observe that the method is insensitive to small perturbations of these parameters, and similar performance is obtained as long as the piecewise structure is preserved.

When $\psi$ is an identify function, the coalition utility becomes additive, leading to the degeneration of GPSV as shown in Theorem B.1. In contrast, the piecewise design of $\psi$ preserves client interaction effects, preventing GPSV from collapsing into the individual marginal contributions. For example, one client may have an overly small projection length, while another client has an overly large one. When evaluated individually, neither receives a high marginal contribution. However, when these two clients form a coalition, their aggregated gradient can fall into the well-aligned and well-scaled region of $\psi$, yielding a high coalition utility. In this case, the SV assigns positive credit to their cooperation, reflecting a synergistic interaction that would be invisible under additive utilities.

### B.3. Details of the GrayEnum Algorithm

We present the detailed steps of the GrayEnum algorithm in Algorithm 2. Given the aggregation weights $\{p_i\}_{i \in S_t}$, the projection length of a coalition $C$, $\lambda_C = \sum_{i \in C} \frac{|\mathcal{D}_i|}{|\mathcal{D}_C|} \lambda_i$, can be reformulated as $\lambda_C = \frac{\sum_{i \in C} p_i \lambda_i}{\sum_{i \in C} p_i}$. With this formulation, GrayEnum maintains two running accumulators $A = \sum_{i \in C} p_i \lambda_i$ and $B = \sum_{i \in C} p_i$ representing the numerator and denominator of $\lambda_C$, respectively. The coalition projection $\lambda_C$ is then obtained directly as $A/B$.

GrayEnum iterates over all $2^{|S_t|}$ subsets using Gray codes, which guarantees that the two consecutive coalitions differ by exactly one client. This property enables incremental updates of $A$ and $B$ by adding or removing the contribution of a single client, rather than recomputing from scratch. As a result, the utility of each coalition can be computed in $\mathcal{O}(1)$ time.

---

**Algorithm 2** GrayEnum: Coalition Utility Computation via Gray Code Enumeration

---

1: **Input:** Gradient projection $\{\lambda_i\}_{i \in S_t}$, aggregation weight $\{p_i\}_{i \in S_t}$

2: $A, B \leftarrow 0, 0$

3: $C_{\text{last}} \leftarrow \emptyset$

4: **for** $g = 0$ **to** $2^{|S_t|} - 1$ **do**

5:      gray $\leftarrow g \oplus (g \gg 1)$     // Bitwise XOR of g and g logically right-shifted by 1.

6:      // Subset of $S_t$ indicated by the bitmask gray.

7:      $C \leftarrow \{S_t[i] \mid 0 \leq i < |S_t| \text{ and } (\text{gray} \& (1 \ll i)) \neq 0\}$

8:      **if** $C_{\text{last}} \neq \emptyset$ **then**

9:          // Symmetric difference of two consecutive subsets.

10:          $Z \leftarrow C \triangle C_{\text{last}}$     // Gray code ensures $|Z| = 1$.

11:          $z \leftarrow$ the only element of $Z$

12:          **if** $z \in C$ **then**

13:              $A \leftarrow A + p_z * \lambda_d$

14:              $B \leftarrow B + p_z$

15:          **else**

16:              $A \leftarrow A - p_z * \lambda_d$

17:              $B \leftarrow B - p_z$

18:          **end if**

19:      **else**

20:          $A \leftarrow \sum_{i \in C} p_i * \lambda_i$

21:          $B \leftarrow \sum_{i \in C} p_i$

22:      **end if**

23:      **if** $B = 0$ **then**

24:          $\lambda_C \leftarrow 0$

25:      **else**

26:          $\lambda_C \leftarrow A/B$

27:      **end if**

28:      $C_{\text{last}} \leftarrow C$

29: **end for**

30: **Return:** $\{\lambda_C\}_{C \subseteq S_t}$

---

## C. Proof of Theorems

### C.1. Preliminaries for Proof

**Assumption C.1** (L-Smoothness). Each local objective function $F_i$ is $L$-smooth. For all $\mathbf{v}, \mathbf{w} \in \mathbb{R}^d$, $F_i(\mathbf{v}) \leq F_i(\mathbf{w}) + (\mathbf{v} - \mathbf{w})^\top \nabla F_i(\mathbf{w}) + \frac{L}{2} \|\mathbf{v} - \mathbf{w}\|_2^2$.

**Assumption C.2** ($\mu$-Strong Convexity). Each local objective function $F_i$ is $\mu$-strongly convex. For all $\mathbf{v}, \mathbf{w} \in \mathbb{R}^d$, $F_i(\mathbf{v}) \geq F_i(\mathbf{w}) + (\mathbf{v} - \mathbf{w})^\top \nabla F_i(\mathbf{w}) + \frac{\mu}{2} \|\mathbf{v} - \mathbf{w}\|_2^2$.

**Assumption C.3** ($\sigma_i^2$-Bounded Gradient Variance). Let $\xi_t^i$ denote a mini-batch uniformly sampled from the local data of client $i$. The variance of the stochastic gradient is bounded. $\mathbb{E} \left\| \nabla F_i(\mathbf{w}_i^t, \xi_t^i) - \nabla F_i(\mathbf{w}_i^t) \right\|^2 \leq \sigma_i^2$ for all $i \in \mathcal{N}$ and $t = 0, \ldots, T - 1$.

**Assumption C.4** ($G^2$-Bounded Gradient Norm). The expected squared norm of the stochastic gradient is uniformly bounded. $\mathbb{E} \left\| \nabla F_i(\mathbf{w}_i^t, \xi_i^t) \right\|^2 \leq G^2$ for all $i \in \mathcal{N}$ and $t = 0, \ldots, T - 1$.

Following Cho et al. (2020), we quantify data heterogeneity using $\rho$ and $\Gamma$.

**Definition C.5** (Global-Local Objective Gap). The gap between the global optimum and the weighted average of local optima is defined as

$$\Gamma \triangleq F^* - \sum_{i \in \mathcal{N}} p_i F_i(\mathbf{w}_i^*) \geq 0. \tag{21}$$

**Definition C.6** (Client Skew). Let $\mathbf{w}'$ denote the model state at evaluation. The client skew of a subset $S$ is defined as

$$\rho(S, \mathbf{w}') = \frac{\mathbb{E}_S[\sum_{i \in S} p_i(F_i(\mathbf{w}') - F_i(\mathbf{w}_i^*))]}{\sum_{i \in \mathcal{N}} p_i(F_i(\mathbf{w}') - F_i(\mathbf{w}_i^*))} \geq 0, \tag{22}$$

with $\bar{\rho} \triangleq \min_{\mathbf{w}'} \rho(S, \mathbf{w}')$ and $\widetilde{\rho} \triangleq \max \rho(S, \mathbf{w}^*)$ to facilitate analysis.

Using Assumptions C.1 and C.2, and letting $\mathbf{w}'_{\min} = \arg\min_{\mathbf{w}'} \rho(S, \mathbf{w}')$, we immediately obtain the following relationships between $\rho$ from Cho et al. (2020) and our proposed gradient based variant $\rho_\nabla$:

$$\bar{\rho}_\nabla = \frac{\mathbb{E}_S[\sum_{i \in S} p_i \|\nabla F_i(\mathbf{w}'_{\min})\|^2]}{2L \sum_{i \in \mathcal{N}} p_i(F_i(\mathbf{w}'_{\min}) - F_i(\mathbf{w}_i^*))} \leq \bar{\rho}, \tag{23}$$

$$\tilde{\rho}_\nabla = \frac{\mathbb{E}_S[\sum_{i \in S} p_i \|\nabla F_i(\mathbf{w}^*)\|^2]}{2\mu \sum_{i \in \mathcal{N}} p_i(F_i(\mathbf{w}^*) - F_i(\mathbf{w}_i^*))} \geq \tilde{\rho}. \tag{24}$$

Finally, We introduce a useful lemma:

**Lemma C.7.** *Assume Assumption C.1 holds, we have that*

$$\|\nabla F_i(\mathbf{w}_i)\|^2 \leq 2L(F_i(\mathbf{w}_i) - F_i(\mathbf{w}_i^*)). \tag{25}$$

## C.2. Proof of Theorem 4.2

With $\mathbf{g}_i^t = \nabla F_i(w^t, \xi_i^t)$ and $\overline{\mathbf{g}}_i^t = \nabla F_i(w^t)$, we have

$$\|\mathbf{w}^{t+1} - \mathbf{w}^*\|^2 = \|\mathbf{w}^t - \eta_t \mathbf{g}_t - \mathbf{w}^* - \eta_t \overline{\mathbf{g}}_t + \eta_t \overline{\mathbf{g}}_t\|^2$$
$$= \underbrace{\|\mathbf{w}^t - \mathbf{w}^* - \eta_t \overline{\mathbf{g}}_t\|^2}_{A} + \underbrace{2\eta_t \langle \mathbf{w}^t - \mathbf{w}^* - \eta_t \overline{\mathbf{g}}_t, \overline{\mathbf{g}}_t - \mathbf{g}_t \rangle}_{B} + \underbrace{\eta_t^2 \|\overline{\mathbf{g}}_t - \mathbf{g}_t\|^2}_{C}. \tag{26}$$

We further decompose $A$ as:

$$A = \|\mathbf{w}^t - \mathbf{w}^*\|^2 + \underbrace{\eta_t^2 \|\overline{\mathbf{g}}_t\|^2}_{A_1} + \underbrace{\left(-2\eta_t \langle \mathbf{w}^t - \mathbf{w}^*, \overline{\mathbf{g}}_t \rangle\right)}_{A_2}. \tag{27}$$

From the L-smoothness of $F_i$, it follows that

$$\|\nabla F_i(\mathbf{w}_i^t)\|^2 \leq 2L(F_i(\mathbf{w}_i^t) - F_i(\mathbf{w}_i^*)). \tag{28}$$

Consequently, $A_1$ can be bounded as

$$A_1 \leq \eta_t^2 \sum_{i \in S_t} p_i \left\|\nabla F_i(\mathbf{w}_i^t)\right\|^2 \tag{29}$$

$$\leq 2L\eta_t^2 \sum_{i \in S_t} p_i \left(F_i(\mathbf{w}_i^t) - F_i(\mathbf{w}_i^*)\right), \tag{30}$$

where Eq. 29 is follows from Jensen's inequality.

We next bound $A_2$

$$A_2 = -2\eta_t \sum_{i \in S_t} p_i \left[ \langle \mathbf{w}^t - \mathbf{w}_i^t, \nabla F_i(\mathbf{w}_i^t) \rangle + \langle \mathbf{w}_i^t - \mathbf{w}^*, \nabla F_i(\mathbf{w}_i^t) \rangle \right]$$

$$\leq \sum_{i \in S_t} p_i \left\| \mathbf{w}^t - \mathbf{w}_i^t \right\|^2 + \eta_t^2 \sum_{i \in S_t} p_i \left\| \nabla F_i(\mathbf{w}_i^t) \right\|^2 + 2\eta_t \sum_{i \in S_t} p_i \left\langle \mathbf{w}^* - \mathbf{w}_i^t, \nabla F_i(\mathbf{w}_i^t) \right\rangle \tag{31}$$

$$\leq \sum_{i \in S_t} p_i \left\| \mathbf{w}^t - \mathbf{w}_i^t \right\|^2 + 2L\eta_t^2 \sum_{i \in S_t} p_i \left( F_i(\mathbf{w}_i^t) - F_i(\mathbf{w}_i^*) \right) + 2\eta_t \sum_{i \in S_t} p_i \left\langle \mathbf{w}^* - \mathbf{w}_i^t, \nabla F_i(\mathbf{w}_i^t) \right\rangle \tag{32}$$

$$\leq \sum_{i \in S_t} p_i \left\| \mathbf{w}^t - \mathbf{w}_i^t \right\|^2 + 2L\eta_t^2 \sum_{i \in S_t} p_i \left( F_i(\mathbf{w}_i^t) - F_i(\mathbf{w}_i^*) \right) - 2\eta_t \sum_{i \in S_t} p_i \left( F_i(\mathbf{w}_i^t) - F_i\left(\mathbf{w}^*\right) \right) - \eta_t \mu \left\| \overline{\mathbf{w}}^t - \mathbf{w}^* \right\|^2, \tag{33}$$

where Eq. 31 follows from Young's inequality, Eq. 32 follows from Lemma C.7, and Eq. 33 follows from $\mu$-convexity. With $A_1$ and $A_2$, we bound $A$ as:

$$A \leq (1 - \eta_t \mu)\|\mathbf{w}^t - \mathbf{w}^*\|^2 + \sum_{i \in S_t} p_i \|\mathbf{w}^t - \mathbf{w}_i^t\|^2 + \underbrace{4L\eta_t^2 \sum_{i \in S_t} p_i \left( F_i(\mathbf{w}_i^t) - F_i(\mathbf{w}_i^*) \right) - 2\eta_t \sum_{i \in S_t} p_i \left( F_i(\mathbf{w}_i^t) - F_i\left(\mathbf{w}^*\right) \right)}_{D}. \tag{34}$$

We next bound $D$ with $\eta_t < \frac{1}{4L}$ and $\delta_t = 2\eta_t(1 - 2L\eta_t)$

$$D = \sum_{i \in S_t} p_i \left[ 4L\eta_t^2 F_i(\mathbf{w}_i^t) - 2\eta_t F_i(\mathbf{w}_i^t) - \left( 4L\eta_t^2 F_i(\mathbf{w}_i^*) - 2\eta_t F_i(\mathbf{w}_i^*) \right) \right] + 2\eta_t \sum_{i \in S_t} p_i \left( F_i\left(\mathbf{w}^*\right) - F_i(\mathbf{w}_i^*) \right)$$

$$= -2\eta_t(1 - 2L\eta_t) \sum_{i \in S_t} p_i \left( F_i(\mathbf{w}_i^t) - F_i(\mathbf{w}_i^*) \right) + 2\eta_t \sum_{i \in S_t} p_i \left( F_i\left(\mathbf{w}^*\right) - F_i(\mathbf{w}_i^*) \right)$$

$$= -\delta_t \underbrace{\sum_{i \in S_t} p_i \left( F_i(\mathbf{w}_i^t) - F_i(\mathbf{w}_i^*) \right)}_{H} + 2\eta_t \sum_{i \in S_t} p_i \left( F_i\left(\mathbf{w}^*\right) - F_i(\mathbf{w}_i^*) \right). \tag{35}$$

We next bound $H$ as:

$$H = \sum_{i \in S_t} p_i \left[ F_i(\mathbf{w}_i^t) - F_i(\mathbf{w}^t) + F_i(\mathbf{w}^t) - F_i(\mathbf{w}_i^*) \right]$$

$$\geq \sum_{i \in S_t} p_i \left[ \langle \mathbf{w}_i^t - \mathbf{w}^t, \nabla F_i(\mathbf{w}^t) \rangle + F_i(\mathbf{w}^t) - F_i(\mathbf{w}_i^*) \right] \tag{36}$$

$$\geq \sum_{i \in S_t} p_i \left[ -\frac{1}{2\eta_t} \left\| \mathbf{w}^t - \mathbf{w}_i^t \right\|^2 - \frac{\eta_t}{2} \left\| \nabla F_i(\mathbf{w}^t) \right\|^2 + F_i(\mathbf{w}^t) - F_i(\mathbf{w}_i^*) \right] \tag{37}$$

$$\geq (1 - \eta_t L) \sum_{i \in S_t} p_i(F_i(\mathbf{w}^t) - F_i(\mathbf{w}_i^*)) - \sum_{i \in S_t} p_i(\frac{1}{2\eta_t} \left\| \mathbf{w}^t - \mathbf{w}_i^t \right\|^2), \tag{38}$$

where Eq. 36 follows from the $\mu$-convexity, Eq. 37 follows from Young's inequality and Eq. 38 follows from Eq. 28.

With $H$, we can further bound $D$ as:

$$
\begin{aligned}
D &\leq -\delta_t(1-\eta_t L)\sum_{i\in S_t} p_i(F_i(\mathbf{w}^t) - F_i(\mathbf{w}_i^*)) + \sum_{i\in S_t} p_i(\frac{\delta_t}{2\eta_t}\left\|\mathbf{w}^t - \mathbf{w}_i^t\right\|^2) + 2\eta_t\sum_{i\in S_t} p_i\left(F_i\left(\mathbf{w}^*\right) - F_i(\mathbf{w}_i^*)\right) \\
&\leq \sum_{i\in S_t} p_i\left\|\mathbf{w}^t - \mathbf{w}_i^t\right\|^2 - \delta_t(1-\eta_t L)\sum_{i\in S_t} p_i(F_i(\mathbf{w}^t) - F_i(\mathbf{w}_i^*)) + 2\eta_t\sum_{i\in S_t} p_i\left(F_i\left(\mathbf{w}^*\right) - F_i(\mathbf{w}_i^*)\right) \quad\quad (39) \\
&= \sum_{i\in S_t} p_i\left\|\mathbf{w}^t - \mathbf{w}_i^t\right\|^2 - \delta_t(1-\eta_t L)\sum_{i\in S_t} p_i\rho(\pi, \mathbf{w}^{(K\lfloor t/K\rfloor)}), \mathbf{w}^t(F(\mathbf{w}^t) - \sum_{i\in S_t} p_i F_i(\mathbf{w}_i^*)) \\
&\quad + 2\eta_t\sum_{i\in S_t} p_i\rho(S(\pi, \mathbf{w}^{(K\lfloor t/K\rfloor)}), \mathbf{w}^*)(F^* - \sum_{i\in S_t} p_i F_i(\mathbf{w}_i^*)) \quad\quad (40) \\
&\leq \sum_{i\in S_t} p_i\left\|\mathbf{w}^t - \mathbf{w}_i^t\right\|^2 - \delta_t(1-\eta_t L)\bar\rho(F(\mathbf{w}^t) - \sum_{i\in S_t} p_i F_i(\mathbf{w}_i^*)) + 2\eta_t\tilde\rho\Gamma \quad\quad (41) \\
&\leq \sum_{i\in S_t} p_i\left\|\mathbf{w}^t - \mathbf{w}_i^t\right\|^2 \underbrace{-\delta_t(1-\eta_t L)\bar\rho_\nabla(F(\mathbf{w}^t) - \sum_{i\in S_t} p_i F_i(\mathbf{w}_i^*))}_{I} + 2\eta_t\tilde\rho_\nabla\Gamma, \quad\quad (42)
\end{aligned}
$$

where Eq. 39 follows from $1 - 2L\eta_t < 1$, Eq. 40 and Eq. 41 follows from Def. C.6 and Def. C.5, and Eq. 42 follows from Eq. 23 and Eq. 24.

We next bound $I$ as:

$$
\begin{aligned}
I &= -\delta_t(1-\eta_t L)\bar\rho_\nabla(F_i(\mathbf{w}^t) - F^*) - \delta_t(1-\eta_t L)\bar\rho_\nabla(F^* - \sum_{i\in S_t} p_i F_i(\mathbf{w}_i^*)) \\
&= -\delta_t(1-\eta_t L)\bar\rho_\nabla(F(\mathbf{w}^t) - F^*) - \delta_t(1-\eta_t L)\bar\rho_\nabla\Gamma \quad\quad (43) \\
&\leq -\frac{\delta_t(1-\eta_t L)\mu\bar\rho_\nabla}{2}\|\mathbf{w}^t - \mathbf{w}^*\|^2 - \delta_t(1-\eta_t L)\bar\rho_\nabla\Gamma \quad\quad (44) \\
&\leq -\frac{3\eta_t\mu\bar\rho_\nabla}{8}\|\mathbf{w}^t - \mathbf{w}^*\|^2 - 2\eta_t(1-2\eta_t L)(1-\eta_t L)\bar\rho_\nabla\Gamma \quad\quad (45) \\
&\leq -\frac{3\eta_t\mu\bar\rho_\nabla}{8}\|\mathbf{w}^t - \mathbf{w}^*\|^2 - 2\eta_t\Gamma\bar\rho_\nabla + 6\eta_t^2 L\Gamma\bar\rho_\nabla, \quad\quad (46)
\end{aligned}
$$

where Eq. 43 follows from Def. C.5, Eq. 44 follows from $u$-convexity, and Eq. 45 follows from $(1 - 2\eta_t L)(1 - \eta_t L) \geq \frac{3}{8}$ with $\eta_t < \frac{1}{4L}$ and Eq. 46 follows from $(1 - 2\eta_t L)(1 - \eta_t L) \geq 1 - 3\eta_t L$.

We further bound $D$ as:

$$
D \leq \sum_{i\in S_t} p_i\left\|\mathbf{w}_i^t - \mathbf{w}^t\right\|^2 - \frac{3\eta_t\mu\bar\rho_\nabla}{8}\|\mathbf{w}^t - \mathbf{w}^*\|^2 + 2\eta_t\Gamma(\tilde\rho_\nabla - \bar\rho_\nabla) + 6\eta_t^2 L\Gamma\bar\rho_\nabla. \quad\quad (47)
$$

We further bound $A$ as:

$$
A1 \leq \left[(1 - \eta_t\mu(1 + \frac{3\bar\rho_\nabla}{8}))\right]\|\mathbf{w}^t - \mathbf{w}^*\|^2 + 2\underbrace{\sum_{i\in S_t} p_i\left\|\mathbf{w}_i^t - \mathbf{w}^t\right\|^2}_{J} + 2\eta_t\Gamma(\tilde\rho_\nabla - \bar\rho_\nabla) + 6\eta_t^2 L\Gamma\bar\rho_\nabla.
$$

We next bound $J$ as:

$$\sum_{i \in S_t} p_i \|\mathbf{w}^t - \mathbf{w}_i^t\|^2 = \sum_{i \in S_t} p_i \|\sum_{j \in S_t} p_j(\mathbf{w}_j^t - \mathbf{w}_i^t)\|^2$$

$$\leq \sum_{i,j \in S_t} p_i p_j \|\mathbf{w}_j^t - \mathbf{w}_i^t\|^2 \tag{48}$$

$$= \sum_{i,j \in S_t} p_i p_j \|(\mathbf{w}_j^t - \mathbf{w}^{t-1}) - (\mathbf{w}_i^t - \mathbf{w}^{t-1})\|^2$$

$$\leq 2 \sum_{i,j \in S_t} p_i p_j \left( \|\mathbf{w}_j^t - \mathbf{w}^{t-1}\|^2 + \|\mathbf{w}_i^t - \mathbf{w}^{t-1}\|^2 \right) \tag{49}$$

$$= 2 \sum_{i,j \in S_t} p_i p_j \left[ \|\sum_{k=0}^{K-1} \eta_{t-1} \nabla F_j(\mathbf{w}_j^{t-1,k}, \xi_j^t)\|^2 + \|\sum_{k=0}^{K-1} \eta_{t-1} \nabla F_i(\mathbf{w}_i^{t-1,k}, \xi_i^k)\|^2 \right]$$

$$\leq 2\eta_{t-1}^2 K \sum_{i,j \in S_t} \sum_{k=0}^{K-1} p_i p_j \left[ \|\nabla F_j(\mathbf{w}_j^{t-1,k}, \xi_j^t)\|^2 + \|\nabla F_i(\mathbf{w}_i^{t-1,k}, \xi_i^k)\|^2 \right] \tag{50}$$

$$\leq 4\eta_{t-1}^2 K \sum_{i,j \in S_t} \sum_{k=0}^{K-1} p_i p_j G^2 \tag{51}$$

$$\leq 16\eta_t^2 K^2 G^2, \tag{52}$$

where Eq. 48 and Eq. 50 follow from Jensen's inequality, Eq. 49 follows from Cauchy–Schwarz inequality and Eq. 49 follows from a non-increasing learning rate with $\eta_{t-1} < 2\eta_t$.

So finally we bound $A$ as:

$$A \leq \left[ (1 - \eta_t \mu (1 + \frac{3\bar{\rho}_\nabla}{8})) \right] \|\mathbf{w}^t - \mathbf{w}^*\|^2 + 32\eta_t^2 K^2 G^2 + 2\eta_t \Gamma(\tilde{\rho}_\nabla - \bar{\rho}_\nabla) + 6\eta_t^2 L\Gamma\bar{\rho}_\nabla. \tag{53}$$

With $E[\mathbf{g}_t] = \bar{\mathbf{g}}_t$

$$E[B] = 0. \tag{54}$$

With Assumption C.4, $C$ can be bounded as:

$$E[C] \leq \eta_t^2 \sigma^2. \tag{55}$$

Combining the above, Eq. 26 yields

$$\mathbb{E}[\|\mathbf{w}^{t+1} - \mathbf{w}^*\|^2] \leq \left[ 1 - \eta_t \mu \left( 1 + \frac{3\bar{\rho}_\nabla}{8} \right) \right] \mathbb{E}[\|\mathbf{w}^t - \mathbf{w}^*\|^2] + \eta_t^2 \left( 32K^2 G^2 + \sigma^2 + 6L\Gamma\bar{\rho}_\nabla \right)$$
$$+ 2\eta_t \Gamma(\tilde{\rho}_\nabla - \bar{\rho}_\nabla). \tag{56}$$

With $\Delta_{t+1} = \mathbb{E}[\|\mathbf{w}^{t+1} - \mathbf{w}^*\|^2]$, $X = 1 + \frac{3\bar{\rho}_\nabla}{8}$, $Y = 32K^2 G^2 + \sigma^2 + 6L\Gamma\bar{\rho}_\nabla$, and $Z = 2\Gamma(\tilde{\rho}_\nabla - \bar{\rho}_\nabla)$, and assuming $\Delta_t \leq \frac{V}{t+\gamma}$ with $\eta_t = \frac{\beta}{t+\gamma}, \gamma > 0$ and $\beta > \frac{16}{\mu(8+3\bar{\rho}_\nabla)}$, we have

$$\Delta_{t+1} \leq (1 - \eta_t \mu X)\Delta_t + \eta_t^2 Y + \eta_t Z$$
$$= \left( 1 - \frac{\beta\mu X}{t+\gamma} \right) \cdot \frac{V}{t+\gamma} + \left( \frac{\beta}{t+\gamma} \right)^2 Y + \frac{\beta Z}{t+\gamma}$$
$$= \frac{V + \beta Z}{t+\gamma} + \frac{-\beta\mu XV + \beta^2 Y}{(t+\gamma)^2}. \tag{57}$$

To ensure $\Delta_{t+1} \leq \frac{V}{t+\gamma+1}$, we require

$$\frac{V + \beta Z}{t + \gamma} + \frac{-\beta \mu X V + \beta^2 Y}{(t + \gamma)^2} \leq \frac{V}{t + \gamma} - \frac{V}{(t + \gamma)(t + \gamma + 1)}. \tag{58}$$

With $\beta \mu X - \frac{t+\gamma}{t+\gamma+1} \geq 1$, this holds if $V$ satisfies

$$V \geq \frac{\beta Z(t + \gamma) + \beta^2 Y}{\beta \mu X - \frac{t+\gamma}{t+\gamma+1}}$$
$$V \geq \beta Z(t + \gamma) + \beta^2 Y \tag{59}$$

and the base case $V \geq \gamma \|\mathbf{w}^0 - \mathbf{w}^*\|^2$ is satisfied. Therefore, it suffices to choose $V = \beta^2 Y + \beta Z(t + \gamma) + \gamma \|\mathbf{w}^0 - \mathbf{w}^*\|^2$

By $L$-smoothness, we finally obtain

$$\mathbb{E}[F(\mathbf{w}^{(t)})] - F^* \leq \frac{L}{2}\Delta_t$$
$$= \frac{L}{2(t + \gamma)} \left( \gamma \|\mathbf{w}^0 - \mathbf{w}^*\|^2 + 32\beta^2 K^2 G^2 + \beta^2 \sigma^2 + 6\beta^2 L\Gamma\bar{\rho}_\nabla \right) + \beta L\Gamma(\widetilde{\rho}_\nabla - \bar{\rho}_\nabla). \tag{60}$$

### C.3. Proof of Theorem 4.5

After selection with retention ratio $r$, the weight $p_i$ of each client is scaled to $\frac{p_i}{r}$, and the lower bound of Theorem 4.5 is:

$$\frac{\rho_\nabla^+}{\rho_\nabla} = \sum_{i \in S^+} \frac{p_i}{r} \frac{\|\mathbf{g_i}\|^2}{\|\mathbf{g}\|^2}$$
$$\geq \frac{1}{\|\mathbf{g}\|^2} \sum_{i \in S^+} \frac{p_i}{r} \left( \|\mathbf{g}_i\| \cos \theta_i \right)^2 \tag{61}$$
$$\geq \frac{1}{\|\mathbf{g}\|^2} \left( \sum_{i \in S^+} \frac{p_i}{r} \|\mathbf{g}_i\| \cos \theta_i \right)^2 \tag{62}$$
$$= \left( \frac{\sum_{i \in S^+} \phi_i}{\sum_{i \in S} \phi_i} + \varepsilon \right)^2, \tag{63}$$

where $\theta_i$ in Eq. 61 is the angle between the $\mathbf{g}_i$ and $\mathbf{g}$, Eq. 62 follows from Jensen inequality, and (63) is due to Assumption 4.3.
The upper bound is:

$$\frac{\rho_\nabla^+}{\rho_\nabla} = \sum_{i \in S^+} \frac{p_i}{r} \frac{\|\mathbf{g_i}\|^2}{\|\mathbf{g}\|^2}$$
$$= \sum_{i \in S^+} \frac{p_i}{r} (\cos \theta_i)^2$$
$$\leq \sum_{i \in S^+} \frac{p_i}{r} \kappa^2 \tag{64}$$
$$= \frac{\kappa^2}{r}, \tag{65}$$

where Eq 64 is due to Assumption 4.4.

