# OpenReview forum: "ShapCCS: Shapley-Driven Client Coreset Selection in Federated Learning"
_ICML.cc/2026/Conference — ICML 2026 regular_

### Official Review · Reviewer_76Zg · 2026-03-12

**Soundness:** 3
**Presentation:** 3
**Significance:** 3
**Originality:** 3
**Overall Recommendation:** 4
**Confidence:** 5

**Summary:**

This paper proposes ShapCCS, a Shapley-driven client coreset selection framework for federated learning that aims to reduce computation and communication overhead. The authors introduce the GPSV to evaluate client contributions by considering both the directional alignment and magnitude of client gradients with respect to the global update. Based on GPSV, the proposed method selects high-value clients and excludes fragmented ones while optionally applying data-level coreset selection within retained clients. Experiments on several benchmark datasets demonstrate that ShapCCS improves model performance and robustness while reducing training cost in federated learning settings.

**Compliance With Llm Reviewing Policy:**

Affirmed.

**Final Justification:**

This paper proposes ShapCCS, a Shapley-driven client coreset selection framework for federated learning. In their rebuttals, authors respond to most of my concerns with further clarification and extra experiment results. Although there exist some concerns about the reasonability of Assumptions 4.3-4.4, I think the core idea of this paper contributes to the community. I would like to raise my original score.

**Key Questions For Authors:**

See weakness

**Strengths And Weaknesses:**

Strength:
1. The paper introduces a new client-level perspective for coreset selection.
2. It proposes a novel Shapley-based contribution estimation method.

Weakness:
1. ShapCCS requires several pretraining rounds to compute GPSV before performing coreset selection. The additional computational overhead introduced by this pretraining stage does not appear to be discussed. Moreover, client contributions may change during training, and a one-time Shapley score may not remain valid over time. It is unclear whether ShapCCS remains effective in such dynamic settings.
2. The datasets used in the experiments are relatively small. The authors are encouraged to evaluate the method on more challenging datasets (e.g., Tiny-ImageNet, ImageNet-1K, and DomainNet).
3. The comparison with existing client selection methods seems limited to S-FedAvg, which is relatively outdated. The authors should include more baselines related to client selection and contribution estimation.
4. Are Assumptions 4.3 and 4.4 supported by existing literature? These assumptions do not appear to be commonly used.
5. The motivation of the paper could be made clearer. In particular, the authors should further explain why gradient projection is a more suitable metric for Shapley value utility compared to other alternatives (e.g., cosine similarity or gradient norm).
6. In the implementation, GPSV is computed using only the last-layer gradients. Why is the last layer chosen? This design may ignore important gradient information from deeper layers of the model.
7. Computing GPSV requires considering gradient projections from other clients, which may introduce potential privacy concerns. The authors should provide clarification on this issue.

---

> ### Author Rebuttal · Authors · 2026-03-31
>
> We sincerely thank the reviewer for the valuable feedback. Below, we address the raised concerns.
>
> **Q1 Efficiency and stability of GPSV**
>
> **Q1.1 Additional overhead**
>
> Most existing coreset selection methods rely on a pretraining stage. For example, [1] requires 40 pretraining rounds. Few works, such as [2], avoid pretraining but require periodic coreset reconstruction, incurring additional overhead. In our experiments, we set the pretraining rounds to $T_{\text{pre}}=0.1T$ for all methods. Therefore, we believe that the pretraining overhead of ShapCCS is reasonable and consistent with common practice in prior methods.
>
> **Q1.2 Stability of one-time SV**
>
> The local dataset of each client is typically fixed during FL training. As a result, client contributions and their SVs tend to remain relatively stable. In Figure C1 (https://anonymous.4open.science/r/ShapCCS-5CBB/C1.pdf), we show GPSV changes between round 10 and 100: (a) 74% of clients have differences below 0.005; (b) 67% have rank changes below 20 and none exceed 70.
>
> **Q2 More challenging datasets**
>
> We did not include larger datasets. In scenarios with many clients and high heterogeneity, FedAvg often suffers from severe convergence issues on challenging datasets. In such settings, introducing coreset selection may further hinder convergence. Nevertheless, ShapCCS consistently outperforms baselines on widely used datasets, including CIFAR‑10, CINIC‑10 and CIFAR‑100, indicating its effectiveness across various datasets.
>
> **Q3 More baselines**
>
> We appreciate the suggestion to include more baselines. However, we would like to clarify that the primary focus of our work is coreset selection rather than client selection or contribution estimation.
>
> **Q3.1 Client selection**
>
> As discussed in Appendix A, client selection and coreset selection reduce computational overhead in fundamentally different ways. They are two distinct tasks and not directly comparable. Therefore, client selection falls outside the scope of this work.
>
> **Q3.2 Contribution estimation**
>
> We agree that other contribution evaluation methods could be used for budget allocation. This in turn highlights the generality of our client coreset selection paradigm. We compare GPSV only with other SV variants since SV is one of the most representive measures for contribution estimation and GPSV is built upon [3]. Moreover, most prior SV–based works typically compare against other SV variants rather than generic evaluation approaches. Therefore, we restrict the comparison of GPSV to SV-based methods.
>
> **Q4 Assumptions 4.3 and 4.4**
>
> Assumptions 4.3 and 4.4 impose boundedness on client gradient skew and are inspired by [3, 4]. Specifically, [4] establishes an error bound between SV and marginal contribution. [3] further adopts marginal contribution as an approximation of SV.
>
> **Q5 Motivation for gradient projection**
>
> In section 2.2, we have discussed accuracy and gradient similarity utility. We further discuss the gradient norm here. The gradient norm alone only reflects magnitude and lacks directional information. A large norm may correspond to an update moving in the opposite direction of the global objective, implying a small contribution. The table below verifies this by comparing performance using different gradient utilities.
> ||70%|50%|30%|
> |-|-|-|-|
> |Similarity|59.75|56.21|50.27|
> |Norm|59.88|53.48|49.19|
> |Projection|63.71|60.17|53.47|
>
> **Q6 Why the last layer is chosen**
>
> We adopt the last layer gradients inspired by [3]. As discussed in Lines 194–200 of the main text, the signals from the last layer are
> 1. Sufficiently informative: this is supported by [3]. The table below also shows that ShapCCS achieves similar accuracy when using gradients from the full, backbone and last layer.
> ||70%|50%|30%|
> |-|-|-|-|
> |Full|60.59|59.06|57.50|
> |Backbone|62.66|59.27|54.25|
> |Last|63.71|60.17|53.47|
> 2. Computationally efficient: last layer gradients reduce computation compared with full layers, especially in deep neural networks.
> 3. More robust: since projection incorporates norm information, using last layer gradients helps mitigate the influence of gradient explosion in deeper layers.
>
> **Q7 Privacy concerns**
>
> We clarify that computing GPSV relies only on information already exchanged in standard FedAvg. The projection computation is performed entirely on the server side using only the uploaded gradients in the current round. On the contrary, we argue that ShapCCS poses less privacy risk than existing FL coreset selection methods. For example, [2] requires the server to hold a validation set, while [1] requires clients to upload class feature centers. Both expose additional information, leading to privacy risks.
>
> [1] FedCS: Coreset Selection for Federated Learning, CVPR 2025
>
> [2] Gradient Coreset for Federated Learning, WACV 2024
>
> [3] Redefining Contributions: Shapley-Driven Federated Learning, IJCAI 2024
>
> [4] Gradient-Driven Rewards to Guarantee Fairness in Collaborative Machine Learning, NeurIPS 2021

---

> > ### Author Rebuttal · Reviewer_76Zg · 2026-04-03
> >
> > Thank you to the authors for the rebuttal. I appreciate the additional clarifications on GPSV stability, the motivation for the projection-based utility, and the privacy implications. These responses are helpful.
> >
> > That said, my main concerns about empirical rigor and the gap between the theory and the practical evaluation remain largely unresolved. I chose option (c) because adequately addressing these issues would require substantial new experiments and analyses, which are difficult to provide during the rebuttal period. In particular, the following concerns remain:
> >
> > 1.The response on pretraining overhead is still insufficient to support the paper’s efficiency claims. Noting that prior methods also use pretraining only shows that this design is reasonable; it does not show that ShapCCS has an acceptable cost-benefit tradeoff. GPSV computation is not free, so matching $T_{pre}$ across methods does not imply comparable actual overhead. The rebuttal also does not report wall-clock time or additional server-side computation cost, so it remains unclear whether ShapCCS still yields net efficiency gains once GPSV computation and pretraining are included. At present, the efficiency argument still seems to rely mainly on post-selection training, without clearly stating whether selection/pretraining overhead is part of the overall evaluation.
> >
> > 2.The lack of experiments on more challenging datasets is still not well justified. The authors argue that large-scale, highly heterogeneous settings are already difficult for FedAvg and that coreset selection may make them even harder. I do not find this sufficient: precisely because these settings are more challenging, it is important to test whether the method remains effective there. This response also weakens the claimed applicability of ShapCCS, since it raises the question of whether the method mainly works on smaller benchmarks or relatively controlled settings. Moreover, the rebuttal provides no alternative evidence for scalability. Even if larger datasets are infeasible, additional results on Tiny-ImageNet, DomainNet, more clients, or a scalability analysis would help.
> >
> > 3.The response on insufficient baselines is not fully convincing. The authors argue that client selection is out of scope, but this is not fully consistent with the paper’s own narrative: the paper repeatedly contrasts client-level coreset selection with client selection and includes S-FedAvg as a baseline. If the paper aims to argue that client-level coreset selection is more appropriate than plain client selection, this should be supported by broader experimental comparisons rather than avoided in rebuttal.
> >
> > In addition, restricting contribution-estimation comparisons to SV-based methods is also not fully sufficient. My concern is not whether GPSV belongs to the SV family, but whether a framework that relies on contribution scores for budget allocation has broader applicability. If so, why not compare against other reasonable contribution metrics? The argument that prior SV papers often compare only to SV variants seems more like convention than justification. If ShapCCS is meant to be a more general client coreset selection framework, it would be stronger to show its behavior under different scoring rules, or at least against a broader set of contribution-evaluation baselines.
> >
> > 4.The response on Assumptions 4.3 and 4.4 does not sufficiently resolve my concern. The cited papers suggest that these assumptions are inspired by prior gradient-based contribution-evaluation work, but they do not directly support the specific assumptions used here. In particular, Xu et al. (2021) studies participant-level contribution approximation based on cosine similarity and gives an individual-level approximation analysis; this is not equivalent to Assumption 4.3, which assumes coalition utility is approximately proportional to aggregated Shapley mass. Likewise, neither Xu et al. (2021) nor Tastan et al. (2024) establishes a retained-client gradient misalignment bound of the form assumed in 4.4. I therefore still view these assumptions as relatively strong and non-standard, and believe they require either clearer justification or empirical evidence showing that they approximately hold in practice.
> >
> > I want to emphasize that the core idea of the paper is interesting and potentially valuable. However, the current version still falls short in empirical completeness and theoretical support. For this reason, I am keeping my score unchanged, as I believe the paper would benefit from a more substantial revision cycle to address the missing evaluations and strengthen its claims.

---

> > > ### Author Response · Authors · 2026-04-04
> > >
> > > We thank the reviewer for the continued feedback. With the additional space and more specific questions in this round, we further address and clarify concerns 1–3.
> > >
> > > **Q1 Pretraining Overhead**
> > >
> > > We first clarify the selection overhead. As emphasized in the paper, the per-coalition complexity of GPSV is $\mathcal{O}(1)$, resulting in minimal overhead. The table below reports the total selection time of all methods on CIFAR-10 with Dir(0.1) and r = 30%.
> > >
> > > ||ShapCCS|CRAIG|EL2N|CCS|Moderate|GCFL|FedCS|
> > > |-|-|-|-|-|-|-|-|
> > > |Time (s)|10|337|21|19|19|1500|42|
> > >
> > > ShapCCS has the lowest overhead among all methods. Interestingly, the selection time of ShapCCS is even lower than the standalone data-level method (Moderate). This is because ShapCCS constructs a client coreset, thereby reducing the number of clients that require data-level selection. When $|S_t|$ scales, GPSV can be efficiently approximated via GTG truncation. See Q1 in Reviewer AQha for details.
> > >
> > > Second, regarding pretraining overhead, all methods perform pretraining on the full dataset, resulting in same costs.  We acknowledge that our original efficiency claim did not account for the overhead of pretraining and selection. In the camera-ready version, we will restrict this claim to the post-selection training stage.
> > >
> > > We further break down the runtime into different components with r = 30%, including GPSV computation, data-level coreset selection, pretraining, and post-selection training:
> > >
> > > ||GPSV (Client-level)|Moderate (Data-level)|Pretraining|Post-Selection|
> > > |-|- |-|-|-|
> > > |Time (s)|3|7|85|318|
> > >
> > > The original training time without selection is 871s. ShapCCS accelerates post-selection training by about 60% and overall training by roughly 53%. The gap between the theoretical 70% data reduction and the observed 60% speedup mainly comes from fixed costs such as model initialization. We note that wall‑clock time may fluctuate due to system factors, while the post‑selection data volume is exactly reduced to 70% of the original.
> > >
> > > **Q2 More Datasets**
> > >
> > > We provide additional results on Tiny-ImageNet with 100 clients and Dir(0.1).
> > >
> > > ||70%|50%|30%|
> > > |-|-|-|-|
> > > |Moderate|28.01|26.60|18.94|
> > > |ShapCCS|33.72|30.52|27.90|
> > >
> > > Although the overall accuracy is relatively low, ShapCCS still achieves a clear margin. Due to time constraints, we were unable to run additional baselines. However, given the consistent improvements across several datasets in the paper and the clear gains shown here, we believe ShapCCS is likely to maintain its advantage. We will include results on Tiny-ImageNet in the camera-ready version.
> > >
> > > **Q3.1 Client Selection Baselines**
> > >
> > > We include two additional client selection baselines and observe results similar to S-FedAvg.
> > >
> > > ||70%|50%|30%|
> > > |-|-|-|-|
> > > |GPFL [1]|57.38|51.81|43.37|
> > > |HiCS-FL [2]|55.77|53.49|47.15|
> > > |S-FedAvg|56.42|50.61|47.24|
> > > |ShapCCS|63.71|60.17|53.47|
> > >
> > > More importantly, we believe there may be some misunderstanding. Our intention in Appendix A is to clarify that client coreset selection and client selection are two different tasks, rather than to claim that the former is superior to the latter. Since S-FedAvg belongs to client selection and also employs SV, we include it only as a reference to help illustrate this distinction, rather than as a competing baseline. We acknowledge that the current presentation may have caused confusion. We are willing to move the results of S-FedAvg from the main text to Appendix A, or remove Appendix A entirely in the camera-ready version.
> > >
> > > **Q3.2 Contribution Estimation Baselines**
> > >
> > > First, we include two additional contribution estimation baselines. As discussed in both the paper and this rebuttal, Shapley value is one of the most widely used and effective approaches for contribution evaluation, but it incurs high computational cost. The results below demonstrate the effectiveness of SV-based evaluation and show that GPSV significantly reduces the computational overhead of SV.
> > >
> > > ||70%|50%|30%|Time (s)|
> > > |-|-|-|-|-|
> > > |Leave-One-Out|59.07|54.63|52.16|142|
> > > |FedVAE [3]|55.77|53.49|47.15|26|
> > > |GPSV|63.71|60.17|53.47|3|
> > >
> > > We agree with the reviewer that the client coreset selection paradigm can be generalized beyond Shapley-based ShapCCS to a broader FL framework that allows more diverse contribution evaluation methods. However, we view this as an extension strength rather than a limitation of the current paper that focuses on Shapley value.
> > >
> > > [1] GPFL: A Gradient-Projection-Based Client Selection Framework for Efficient Federated Learning, IEEE Internet of Things Journal, 2025
> > >
> > > [2] Heterogeneity-Guided Client Sampling: Towards Fast and Efficient Non-IID Federated Learning, NeurIPS 2024
> > >
> > > [3] FedAVE: Adaptive data value evaluation framework for collaborative fairness in federated learning, Neurocomputing 2024

---

### Official Review · Reviewer_AQha · 2026-03-12

**Soundness:** 4
**Presentation:** 3
**Significance:** 3
**Originality:** 3
**Overall Recommendation:** 4
**Confidence:** 3

**Summary:**

This paper talks about the problem of coreset selection in federated learning. Usual approaches typically select the coreset at the data level, i.e. for each client, they select a subset of their data to form the coreset. However, this can lead to fragmented clients (if they had very few data points to begin with). The authors propose ShapCCS which instead of selecting the coreset at the data level, chooses the clients themselves that contribute the most using a "gradient projection Shapley value" (GPSV) to measure contribution. This is based on projecting the gradient updates made by a client onto the full gradient and using how closely it aligns with the global gradient as the metric. The algorithm they propose can compute per-coalition value scores in constant time. They further use data level coreset selection to further improve the efficiency.

The proposed method reduces the computation and communication cost of training due to the smaller client set participating in training and less data being processed. The paper also provides a convergence analysis for the proposed method and conducts experiments to evaluate it.

**Compliance With Llm Reviewing Policy:**

Affirmed.

**Final Justification:**

The paper proposes ShapCCS a coreset selection strategy based on Gradient Projection Shapley Value which reduces per coalition utility function cost to $\mathcal{O}(1)$. This method reduces both the computation and communication costs due to the smaller number of clients participating.

In their rebuttal, the authors have clarified that their proposed method is compatible with estimation methods such as GTG and have conducted additional experiments for the same. They have also addressed the presentation issues I have mentioned. Therefore, I am satisfied with their rebuttal and will be maintaining my current score.

**Key Questions For Authors:**

1) The GrayEnum algorithm, while it has a $\mathcal{O}(1)$ per coalition cost, does go through each subset incurring a cost of $\mathcal{O}(2^n)$. How does this scale in practice with the number of clients? Are there any approaches you
   can take to reduce the number of subsets you have to go through?

2) There is an interesting line of work on identifying helpful or informative clients in federated learning that is related to multitask peer prediction. I think exploring this might be an interesting direction.

**Limitations:**

yes

**Strengths And Weaknesses:**

- The GPSV algorithm achieves a $\mathcal{O}(1)$ per coalition utility function evaluation complexity which is a huge improvement compared to other methods. Additionally, it has a reduction in communication cost due to the smaller number of clients participating.
- Experimental results show that the proposed method performs well.
- A brief section introducing Shapley value mathematically would be helpful for readers who are not familiar with it.
- It would be helpful to introduce the notation in section 3.2 more clearly. For example, $g_C$
- At several places, the paper mentions GradEnum. I believe that is a typo and should be changed to GrayEnum.

---

> ### Author Rebuttal · Authors · 2026-03-31
>
> We sincerely thank the reviewer for the valuable feedback. Below, we address the raised concerns.
>
> **Weakness**
>
> In the camera-ready version, we will add a brief introduction to the Shapley value as follows:
>
> > Given a set of players $S$ and a value function $\nu(\cdot)$, the Shapley value (SV) for player $i$ is defined as the average marginal contribution of $i$ across all possible permutations of players. An equivalent expression is:
> > $$
> > SV_i=\frac{1}{|S|}\sum_{C\subseteq S\setminus \{i\}}\frac{\nu(C\cup\{i\})-\nu(C)}{{\binom{|S|-1}{|C |}}}.
> > $$
> > The Shapley value satisfies four desirable fairness properties: efficiency (total utility is fully distributed), symmetry (players with equal contributions receive equal SVs), linearity (SV is additive across games), and the null player  property (players with no contribution receive zero SV).
>
> We also agree with the reviewer that the notation in Section 3.2, such as $\mathbf{g}_C$, is not explained clearly. We will add a short explanation to improve readability. In addition, we appreciate the reviewer for catching the typo: “GradEnum” should indeed be “GrayEnum”, and we will correct this throughout the manuscript.
>
> **Q1 Scale of selected client set $|S_t|$**
>
> We thank the reviewer for this thoughtful question. We agree that computing exact GPSV requires evaluating all $2^{|S_t|}$ subsets, which becomes a bottleneck as $|S_t|$ grows. However, as discussed in the paper, GPSV is fully compatible with truncated estimation methods such as GTG [1], which provide an efficient approximation while preserving accuracy.
>
> Specifically, GTG [1] adopts two truncation strategies: a between-round truncation that skips rounds with minimal accuracy gain, and a within-round truncation that stops a permutation early when the marginal utility gain becomes negligible. Unlike accuracy, a stable gradient projection across rounds still indicates an ongoing training progress. Therefore, we only employ within-round truncation in Monte Carlo sampling with 1000 sampled permutations, yielding $1000 \times |S_t|$ evaluated subsets in each round.
>
> We conduct additional experiments comparing exact GPSV with GTG-estimated GPSV. Detailed results can be found at  https://anonymous.4open.science/r/ShapCCS-5CBB/B1.pdf. As shown below, the cost of computing exact GPSV grows exponentially. When $|S_t|=50$, computing the exact SV becomes practically infeasible due to the exponential explosion of $|S_t|$. In comparison, GTG-estimated GPSV remains nearly linear in time because the number of sampled permutations is fixed. Once $|S_t|>20$, the time spent on exact SV computation exceeds the model training time. Therefore, we recommend switching to GTG-based estimation beyond this point.
>
> |$\|S_t\|$|Exact Time (s)|GTG-Est. Time (s)|
> |-|-|-|
> |10|0.01|0.31|
> |15|0.34|0.46|
> |20|16.13|0.67|
> |50|N/A|1.64|
> |100|N/A|3.41|
>
> We further compare the performance of ShapCCS when using exact GPSV and GTG-estimated GPSV. As shown below, the performance gap remains within 3%, indicating that the coreset quality is only marginally affected by the estimation.
>
> |$\|S_t\|$|Exact Accuracy (%)|GTG-Est. Accuracy (%)|
> |-|-|-|
> |10|58.06|58.44|
> |15|61.33|58.37|
> |20|55.21|55.09|
> |50|N/A|50.78|
> |100|N/A|51.11|
>
> Overall, these results suggest that exact GPSV computation becomes a bottleneck as $|S_t|$ grows, while GTG-based truncation provides an effective approximation with little degradation in coreset quality.
>
> **Q2: Relation to multitask peer prediction**
>
> We thank the reviewer for this valuable suggestion. Previously, we are less familar with multitask peer prediction. It aims to identify participants who provide high-quality information by exploiting statistical correlations among participants across multiple tasks. In this sense, multitask peer prediction and GPSV share a related objective of identifying informative participants. Applying ideas from multitask peer prediction to client corset selection could be a meaningful direction for future work.
>
> [1] GTG-Shapley: Efficient and Accurate Participant Contribution Evaluation in Federated Learning, ACM Transactions on Intelligent Systems and Technology, 2022

---

> > ### Author Rebuttal · Reviewer_AQha · 2026-04-04
> >
> > I thank the authors for their detailed rebuttal. The responses address my questions regarding presentation and include additional experiments showing the compatibility of GPSV with estimation methods without worsening accuracy significantly. The rebuttal sufficiently clarifies my questions and I will maintain my current score.

---

### Official Review · Reviewer_LMH7 · 2026-03-15

**Soundness:** 3
**Presentation:** 3
**Significance:** 3
**Originality:** 3
**Overall Recommendation:** 4
**Confidence:** 3

**Summary:**

The paper introduces ShapCCS, a novel client-level coreset selection strategy designed to reduce computational and communication overheads in Federated Learning (FL). The authors identify that existing data-level coreset selection methods enforce uniform retention ratios across clients, ignoring client heterogeneity and introducing "fragmented clients" (clients with marginal data that waste communication resources). To address this, they first propose Gradient Projection Shapley Value (GPSV), an exact Shapley value calculation method with $\mathcal{O}(1)$ per-coalition complexity that captures both the directional alignment and magnitude of client updates without needing a server-held validation set. ShapCCS utilizes GPSV to allocate data retention budgets, excluding fragmented clients and optionally integrating with existing data-level selection methods to compress redundant local data. The authors provide theoretical convergence guarantees and extensive empirical evaluations demonstrating performance gains and noise robustness.

**Compliance With Llm Reviewing Policy:**

Affirmed.

**Key Questions For Authors:**

- In cross-device FL scenarios where the per-round participation cohort $|S_t|$ might scale to hundreds or thousands of clients, $2^{|S_t|}$ exact evaluations become computationally infeasible. At what threshold of $|S_t|$ does the exact GPSV calculation become a bottleneck, and how exactly does the framework perform when falling back on GTG truncation or Monte Carlo sampling? Does the coreset quality degrade significantly?

- Since the efficacy of GPSV is heavily reliant on the quality of the global gradient reference, have you considered using robust aggregation rules (such as Krum, median, or trimmed mean) to compute the reference gradient before evaluating client projections? Would this mitigate the vulnerability to highly coordinated or extreme noise?

- Relying on a static, globally tuned $\tau$ might be difficult in practical FL deployments where the total data volume and distribution are unknown to the server. Is there a viable strategy for dynamically determining or self-adapting $\tau$ based on the variance of the GPSV scores computed during the pretraining phase?

**Limitations:**

yes

**Strengths And Weaknesses:**

- Soundness: The paper provides a solid theoretical foundation, establishing convergence guarantees for ShapCCS (Theorem 4.2) and bounds on the gradient client skew (Theorem 4.5) to prove that the selected coreset maintains a performance comparable to the full dataset. The empirical validation is robust, utilizing multiple datasets (Fashion-MNIST, CINIC-10, CIFAR-10, CIFAR-100) across both IID and non-IID (Dirichlet) data distributions. It also benchmarks against a strong suite of baselines, including both centralized and FL-specific coreset methods.

- Presentation: The concept of "fragmented clients" is intuitively introduced and well-supported by the empirical motivation in Figure 1, making the core problem highly understandable.

- Significance: Unlike previous data-level coreset approaches that only reduce computation, ShapCCS significantly reduces both computation and communication costs by entirely pruning low-contribution clients from the training rounds. Moreover, the framework is modular and can be integrated with existing data-level selection methods

- Originality: While the client-level exclusion is novel, the mechanism relies heavily on a hard threshold $\tau$ for budget redistribution. While empirically effective, threshold-based pruning is a relatively standard heuristic in model compression/selection literature

---

> ### Author Rebuttal · Authors · 2026-03-31
>
> We sincerely thank the reviewer for the valuable feedback. Below, we address the raised concerns.
>
> **Q1 Scale of $|S_t|$**
>
> We agree that computing exact GPSV requires evaluating all $2^{|S_t|}$ subsets, which becomes a bottleneck as $|S_t|$ grows. However, as discussed in the paper, GPSV is fully compatible with truncated estimation methods such as GTG [1], which provide an efficient approximation while preserving accuracy.
>
> We conduct additional experiments comparing exact GPSV with GTG-estimated GPSV. We use 1000 sampled permutations in GTG-based Monte Carlo sampling, yielding $1000 \times |S_t|$ evaluated subsets in each round. Detailed results can be found at  https://anonymous.4open.science/r/ShapCCS-5CBB/A1.pdf. We compare the performance of ShapCCS when using exact GPSV and GTG-estimated GPSV. As shown below, the performance gap remains within 3%, indicating that the coreset quality is only marginally affected by the estimation.
>
> |$\|S_t\|$|Exact Accuracy (%)|GTG-Est. Accuracy (%)|
> |-|-|-|
> |10|58.06|58.44|
> |15|61.33|58.37|
> |20|55.21|55.09|
> |50|N/A|50.78|
> |100|N/A|51.11|
>
> We further compare the computation time. As shown below, the cost of exact GPSV grows exponentially, while GTG-estimated GPSV remains nearly linear in time because the number of sampled permutations is fixed. Once $|S_t|>20$, the time spent on exact SV computation  exceeds the model training time. Therefore, we recommend switching to GTG-based estimation beyond this point.
>
> |$\|S_t\|$|Exact Time (s)|GTG-Est. Time (s)|
> |-|-|-|
> |10|0.01|0.31|
> |15|0.34|0.46|
> |20|16.13|0.67|
> |50|N/A|1.64|
> |100|N/A|3.41|
>
> Overall, these results suggest that exact GPSV computation becomes a bottleneck as $|S_t|$ grows, while GTG-based truncation provides an effective approximation with little degradation in coreset quality.
>
> **Q2 Robust aggregation rules**
>
> We thank the reviewer for the insightful suggestion. In our current design, we do not improve the aggregation rule because the effectiveness of GrayEnum relies on the fact that the global gradient is a linear combination of the client gradients. Aggregation methods such as Krum break this linearity, which would invalidate GrayEnum and thus eliminate its $\mathcal{O}(1)$ complexity advantage.
>
> However, we agree that improving the reference gradient may lead to better performance. To verify this, we calculate SV using the cosine similarity between clients' gradients and the reference gradient obtained via robust rules, which can be viewed as an improved CGSV [2]. To ensure fairness, the gradients obtained via robust aggregation are used only for SV estimation, not for training.
>
> ||70%|50%|30%|
> |-|-|-|-|
> |CGSV (%)|59.8|56.2|50.3|
> |+Krum|60.4|54.8|52.6|
> |+median|60.3|56.4|50.8|
> |+trimmed mean|57.7|57.0|49.5|
>
> The results suggest that a robust reference can be helpful but its benefit is limited, so it does not justify sacrificing $\mathcal{O}(1)$ advantage.  Nevertheless, we believe this is a meaningful direction for future work.
>
> **Q3 Self-adaptive threshold $\tau$**
>
> We address this concern in two aspects. First, as discussed in Section 5.5, ShapCCS is relatively robust to $\tau$. We also introduce ShapCCS\* with $\tau\to\infty$.  It retains the top GPSV clients without requiring any tuning of $\tau$. As reported in Tables 1 and 2, it consistently outperforms all baselines.
>
> Second, we design a self-adaptive threshold $\tilde{\tau}$ based on the statistics of GPSV. Intuitively, $\tilde{\tau}$ should be correlated with the average client data size and adapt to the heterogeneity of client budgets. When the retention ratio $r$ decreases or the coefficient of variation of GPSV ($CV_ϕ$) increases, the budget allocation becomes more uneven, increasing the risk of fragmentation. In such cases, a larger $\tilde{\tau}$ is needed to filter out fragmented clients. Therefore, we design $\tilde{\tau}$ as follows:
>
>
> $$
> \tilde{\tau}=\underbrace{\frac{|\mathcal{D}|}{|\mathcal{N}|}}\_{\text{Avg. Data}} * \underbrace{((1-r)+\text{sigmoid}(CV_{\phi}))}\_{\text{Budget Heterogeneity}}
> $$
>
> Results in the table below show that this adaptive strategy achieves performance comparable to a tuned static threshold.
>
> ||70%|50%|30%|
> |:-:|:-:|:-:|:-:|
> |$\tau$|63.71|60.17|53.47|
> |$\tilde{\tau}$|62.73|59.40|53.61|
>
> We would also like to clarify the concern regarding the server’s access to the total data volume. In FedAvg, the server knows each client’s data size $|\mathcal{D}_i|$ to compute the aggregation weights $p_i^t=|\mathcal{D}\_i|/ \sum\_{j \in S\_t}|\mathcal{D}_j|$. Therefore, ShapCCS does not require any additional information beyond what standard FL methods already require.
>
> [1] GTG-Shapley: Efficient and Accurate Participant Contribution Evaluation in Federated Learning, ACM Transactions on Intelligent Systems and Technology, 2022
>
> [2] Gradient-Driven Rewards to Guarantee Fairness in Collaborative Machine Learning, NeurIPS 2021

---

> > ### Author Rebuttal · Reviewer_LMH7 · 2026-04-04
> >
> > Thank you for the rebuttal, it clarifies my questions and I will maintain my current score.

---

### Decision · Program_Chairs · 2026-04-30

**Decision:**

Accept (regular)

**Comment:**

The reviewers are in general agreement that the paper contains useful contributions after the rebuttal and discussion clarifications.